# Stimulus-dependent representational drift in primary visual cortex

Tyler D. Marks 🔴 [1] & Michael J. Goard 🔴 [1,2,3 ✉]

To produce consistent sensory perception, neurons must maintain stable representations of sensory input. However, neurons in many regions exhibit progressive drift across days. Longitudinal studies have found stable responses to artificial stimuli across sessions in visual areas, but it is unclear whether this stability extends to naturalistic stimuli. We performed chronic 2-photon imaging of mouse V1 populations to directly compare the representational stability of artificial versus naturalistic visual stimuli over weeks. Responses to gratings were highly stable across sessions. However, neural responses to naturalistic movies exhibited progressive representational drift across sessions. Differential drift was present across cortical layers, in inhibitory interneurons, and could not be explained by differential response strength or higher order stimulus statistics. However, representational drift was accompanied by similar differential changes in local population correlation structure. These results suggest representational stability in V1 is stimulus-dependent and may relate to differences in pre-existing circuit architecture of co-tuned neurons.

[1] Neuroscience Research Institute, University of California, Santa Barbara, CA, USA. [2] Department of Molecular, Cellular, and Developmental Biology, University of California, Santa Barbara, CA, USA. [3] Department of Psychological & Brain Sciences, University of California, Santa Barbara, CA, USA. ✉email: michael.goard@lifesci.ucsb.edu

Ongoing experience-dependent and homeostatic synaptic plasticity suggest that neocortical connectivity is in a constant state of flux[1,2]. The potential for ongoing synaptic modification enables animals to rapidly adapt to a changing environment. However, in the face of ongoing plasticity, the cortex must somehow create stable representations of the external world and internal behavioral states in order to reliably represent the external environment and produce behaviors necessary for an animal's survival. To investigate how the brain handles the tradeoff between flexibility and long-term stability, researchers have sought to perform longitudinal measurements of the same neurons over long time periods to measure "representational drift" in neuronal response properties[3].

Many early chronic recording studies used extracellular recordings to track neurons in the motor cortex and hippocampus. These regions serve as suitable targets due to their relevance in producing stable representations for stereotyped behavior, but the results have not been conclusive. Many studies in motor cortices reveal highly stable motor representations[4–6], but others report unstable individual M1 neuron tuning properties underlying stable ensemble-level representations of highly stereotyped motor actions[7–10]. It has often been difficult to draw strong conclusions from electrophysiology experiments due to low sample sizes, electrode drift, ambiguity in neuron identification across sessions, and potential sampling biases. In particular, high spike rate neurons are likely oversampled in blind electrophysiology recordings and may consequently give a biased impression of stability across the population[4,11,12]. Two-photon imaging has several advantages for chronic neural measurements. First, it has granted us insight into the dynamics of subcellular structures[13], such as the finding that sensory experience accelerates dendritic spine instability underlying synaptic turnover in sensory cortex[14] (though see ref. [15]). Second, in vivo 2-photon calcium imaging enables the functional recording of large, structurally identified populations of neurons[16]. Although multiphoton imaging sacrifices temporal resolution afforded by precise spike measurement, its high spatial resolution mitigates sampling biases and reduces ambiguity in neuronal identification during chronic measurements, allowing for longitudinal studies of large neuronal populations[11,17,18].

These developments have produced the opportunity to expand the investigation of cortical stability to other brain areas over longer time periods. Recent studies in the posterior parietal cortex revealed that stable learned associations can be achieved by neuronal populations in the presence of individual neurons whose coding properties continuously drift[19,20]. However, studies in the sensory cortex have frequently found more stable stimulus representation in single neurons[12,16,18,21–25]. While single-neuron representational drift has been theorized to play a role in learning in areas like the motor cortex and hippocampus[3,8,26–28], recent evidence has also demonstrated its presence in odor-evoked responses in the mouse olfactory cortex[29]. It is less clear what purpose would be served by instability in early sensory areas, where the circuitry is tasked with reliably representing external sensory information on a single exposure. Even so, there is substantial trial-to-trial variability in single neurons[30–35], even in the early sensory cortex. The evidence of stable stimulus representation in the face of this variability has led to the suggestion that the functional connectivity of the sensory cortex may grant it robustness to noise while maintaining the ability to undergo experience-dependent plasticity[17].

Almost all chronic recording studies examining stability in the visual cortex have used simple parameterized stimuli, such as oriented drifting gratings, as they present straightforward measurements for determining the stability of individual neuron response properties. These studies are largely in consensus, finding a high degree of stability with respect to orientation tuning, spatial frequency tuning, and size tuning[18,22,36], although responses to such stimuli may be susceptible to longitudinal reduction as a result of repeated stimulus presentation[37]. Previous work has characterized the existence of small subnetworks of highly stimulus-responsive neurons coexisting against a backdrop of relatively unresponsive neurons[38,39], and a recent study reports these highly active neurons to be particularly stable amidst contextual modulation factors[40]. Responses to grating stimuli are also found to be stable following monocular deprivation[25] and even apical or basal dendritic ablation[41]. However, gratings are designed to optimally stimulate the receptive fields of visual cortical neurons and have simple visual statistics compared to an animal's natural visual input. In addition, orientation tuning is widely believed to be "hardwired" early in development[42–45], and iso-oriented neurons exhibit high connectivity[46,47] likely serving to stabilize orientation responses through ongoing Hebbian plasticity[47,48]. Ensembles of neurons driven by naturalistic stimuli are not necessarily iso-tuned, and almost certainly exhibit lower levels of intrinsic connectivity. As a result, there may be considerably more representational drift in response to naturalistic stimuli than to gratings, particularly over long time periods (weeks to months)[49].

Here, we perform chronic 2-photon calcium imaging of thousands of neurons in the primary visual cortex (V1) of awake, head-fixed mice viewing both oriented drifting grating stimuli as well as repeated presentations of a continuous naturalistic movie. We demonstrate stable orientation preference and high stability of grating responses across sessions, consistent with previous work. However, responses to repeated presentations of naturalistic movies exhibited progressive drift across sessions, involving the gain and loss of individual response peaks over the course of several weeks. The stimulus-dependent difference in response stability was true even for neurons exhibiting selectivity to both grating and natural movie stimuli. This representational drift was seen across cortical layers and in both excitatory and inhibitory cell types and could not be explained by response magnitude or eye movements. Finally, we found that representational drift in response to natural stimuli was accompanied by greater drift in the correlation structure of the local neuronal population during natural stimuli than for gratings. Taken together, these results demonstrate that neurons can exhibit different levels of stability to distinct encoded features.

## Results

**Visual cortical neurons exhibit representational drift in response to natural stimuli.** We performed chronic 2-photon imaging to measure single-cell visual responses over many weeks in the primary visual cortex (V1) of awake, head-fixed transgenic mice expressing the calcium indicator GCaMP6s in excitatory neurons (13 fields in 12 mice[50–52]). The mice passively viewed the same visual stimuli on every session (Fig. 1a, b; see "Methods"), starting with a repeated sequence of periodic oriented drifting gratings (passive drifting grating (PDG)), followed by repeated presentations of a 30 s continuous naturalistic movie (MOV). Recordings were performed at 7 ± 1 day intervals for a total of 5–7 weeks. To ensure our selected imaging fields solely contained V1 neurons, we used a widefield microscope to determine visual area boundaries through established retinotopic mapping procedures[53–55] (Fig. 1b, "Methods"). Visual landmarks such as blood vessels were used to identify and align the 2-photon imaging field on a given recording session, and small differences in the alignment of the horizontal plane were corrected in postprocessing using nonrigid image registration software[9]. Somatic regions of interest (ROIs) were defined based on the average

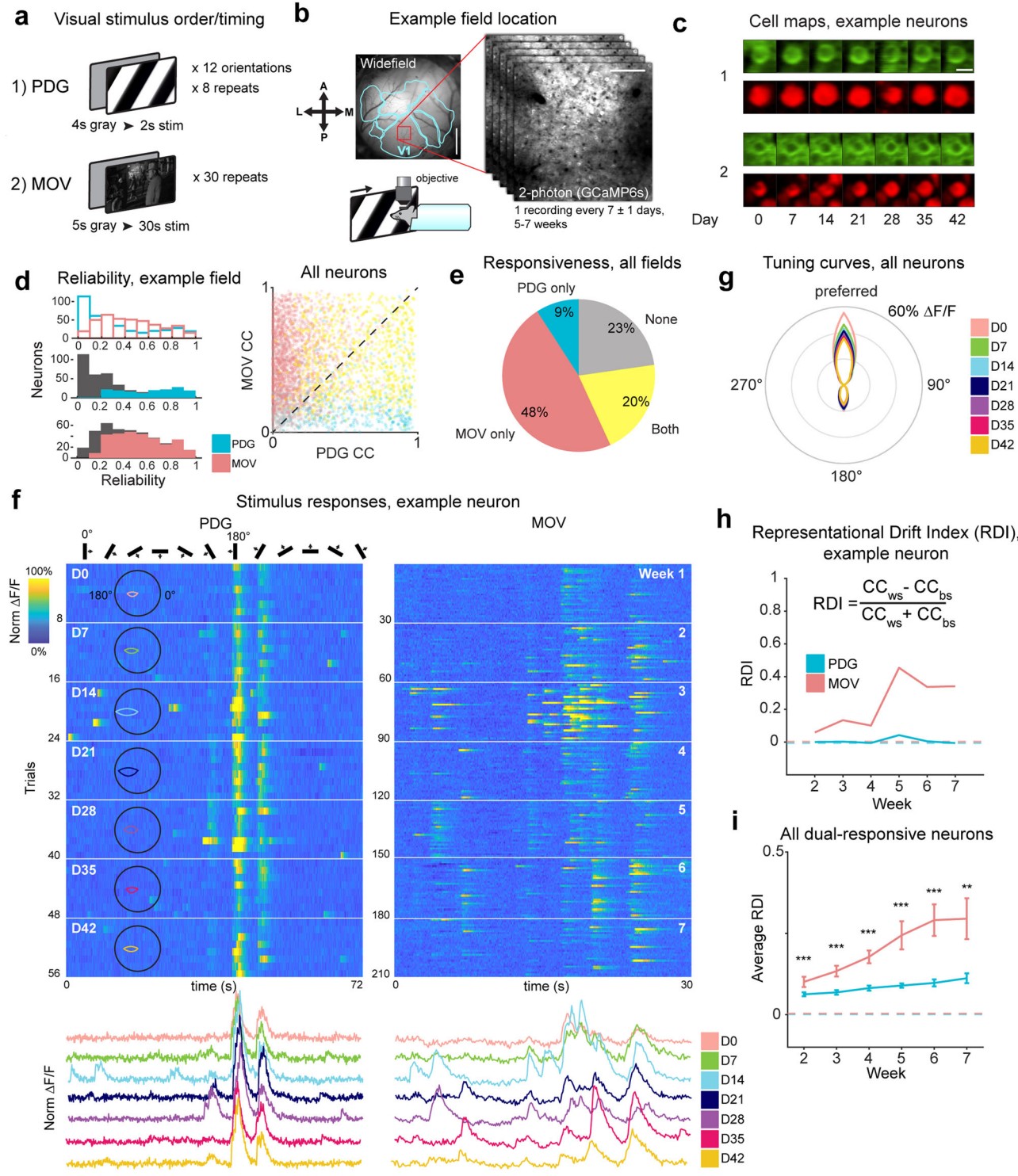

fluorescence maps and average pixelwise activity maps across sessions (see "Methods"). To mitigate experimental artifacts stemming from misalignments of the field in the z-direction (depth) and to ensure unambiguous identification of individual neurons across sessions, we manually inspected each defined ROI across all sessions. We assigned each cell a quality rating describing the neuron's structural robustness and a presence indicator on a session-by-session basis (Fig. 1c, Supplementary Fig. 1[19]). ROIs were considered for analysis only if they met a sufficient quality threshold (quality index ≥ 3; 4143 ROIs out of 5689) and if the neuron was present on all analyzed sessions.

To characterize a neuron's responsiveness to a given visual stimulus, we calculated its "reliability" on every session[56], defined by the Pearson correlation coefficient (CC) of the session-averaged activity of two random halves of trials, iterated many times, and averaged (see "Methods"). Reliability for both stimuli followed a skewed distribution, with a higher number of neurons responding reliably to the MOV stimulus (Fig. 1d). To determine which neurons were visually responsive we tested the actual reliability of the response to each stimulus against a null distribution of reliability calculated with circularly shifted data ("Methods"). We found that while the majority of neurons (68%

**Fig. 1 Chronic 2-photon imaging reveals differential stability of visual responses in single cells. a** Visual stimuli: top screen depicts a drifting grating of one orientation as presented in the passive drifting grating (PDG) stimulus, bottom screen depicts a single frame from the natural movie (MOV) stimulus. PDG is presented first as 8 repeats of a 12 orientation sequence, followed by 30 repeats of MOV. **b** Location of one example recording field in primary visual cortex (V1). Left: widefield fluorescence image of a 4 mm cortical window for one example Emx1-cre × ROSA-tTA × TITL-GCaMP6s mouse, with overlay (light blue) of visual area boundaries determined by retinotopic mapping (see "Methods"); red box indicates the approximate location of 2-photon recordings in V1; scale bar = 1 mm. Right: example average projection of a 2-photon imaging field; scale bar = 100 μm. Bottom: schematic of the head-fixed mouse. **c** Example images of registered cells (see "Methods") from the imaging field in (b) on all recording days. Top row, green: average projection of GCaMP fluorescence channel. Bottom row, red: pixel-wise activity map (see "Methods"). Scale bar = 15 μm. **d** Top left: single-cell reliability distributions for PDG and MOV stimuli on the first recording session for one example mouse. Reliability is defined as the Pearson correlation coefficient (CC) of trial-averaged activity from two halves of the trials. Middle left: PDG reliability distribution; a subset of PDG responsive neurons is colored. Bottom left: MOV reliability distribution; a subset of MOV responsive neurons is colored. Right: each neuron's between-trial CC for PDG vs. MOV, for neurons present on the reference session across all mice ($n = 4142$ neurons). Dots are colored by significant responsiveness to stimuli, as in (**e**). **e** Average percentage of neurons significantly responsive to each stimulus (MOV only: 47.7 ± 2.1% sem, PDG only: 9.1 ± 0.9%, both: 20.4 ± 2.3%, none: 22.7 ± 2.2%). **f** Fluorescence traces ($\Delta F/F$) for one example neuron. Trials are concatenated across sessions. Left: responses to the PDG stimulus; overlay: orientation tuning curves for each recording day. Right: responses to the MOV stimulus. White horizontal lines in each heatmap indicate divisions between recording sessions (8 trials per session for PDG, 30 trials per session for MOV). Heatmaps for each stimulus are co-normalized. Below each heatmap are trial-averaged responses colored by session. **g** Orientation tuning curves colored by session averaged across all orientation-tuned neurons in all imaging fields and aligned to 0° based on preferred orientation. Neurons are only included if they are present on a given session and orientation tuned (740, 710, 694, 670, 701, 659, 596 neurons per session 1–7 respectively). **h** Top: representational drift index (RDI) curves for each stimulus for example neuron shown in (**f**); values closer to 0 indicates a more stable response (similar to the first recording session), and closer to 1 indicates greater response drift (see methods); inset: RDI formula: $CC_{WS}$ = within-session correlation coefficient, $CC_{BS}$ = between-session correlation coefficient; dotted line indicates control RDI for this cell, determined using half the trials of the session 1 as the reference and the other half as a test data set (see methods). **i** Average RDI curves across all imaging fields. Values for each imaging field on a given session are calculated by averaging across neurons that are present on that session and visually responsive to both stimuli. The dotted line indicates control RDI, as in (**h**). Error bars are ± sem. Significance markers indicate the comparison of average RDI between stimuli for each session ($n = 824, 808, 793, 830, 761, 698$ neurons from 13, 12, 12, 13, 11, 9 imaging fields for sessions 2–7 respectively; $F_{1,1648} = 13.0$, $p = 3.2 \times 10^{-4}$, $F_{1,1616} = 53.0$, $p = 5.2 \times 10^{-13}$, $F_{1,1586} = 23.9$, $p = 1.1 \times 10^{-6}$, $F_{1,1660} = 12.9$, $p = 3.4 \times 10^{-4}$, $F_{1,1522} = 19.9$, $p = 8.6 \times 10^{-6}$, $F_{1,1396} = 8.0$, $p = 4.8 \times 10^{-3}$ for sessions 2–7 respectively; two-tailed F-test using a linear mixed-effects model, fixed effect for stimulus, random effect for mouse; \*\*$p < 0.01$, \*\*\*$p < 0.001$).

of total) were visually responsive to the MOV stimulus and a smaller proportion (29% of total) were responsive to PDG, an even smaller subset (20% of total) were responsive to both stimuli, which we called "dual responsive" (Fig. 1e). In addition, a neuron's response strength for one of the stimuli was uncorrelated with that of the other (Pearson correlation between neurons' z-scored trial-averaged activity for both stimuli, $r = 0.01$, $p > 0.05$).

Consistent with previous work characterizing the stability of visual responses to gratings[22,23,25,36,40], the PDG responsive neurons we recorded exhibited highly stable orientation-tuned responses (Fig. 1f, g, Supplementary Fig. 2). Orientation tuning curves were strongly aligned across sessions for both single neurons (Supplementary Fig. 2a, b) and for all tuned neurons across all mice (Fig. 1g). Changes in orientation selectivity (OSI) were minimal (4.4 ± 0.3% change from reference session on average across all neurons present and tuned across all sessions) and did not increase as a function of elapsed time (Supplementary Fig. 2c). In the vast majority of cases (96% of neurons tuned on the first session), shifts in orientation preference over 4–6 weeks fell within ±1 orientation in the sequence of presentations (Supplementary Fig. 2d).

Responses to the MOV stimulus, however, revealed striking differences when compared to the PDG responses (Supplementary Fig. 3, Supplementary Fig. 4a, b). Qualitatively, we found single neuron responses to MOV to be volatile across sessions, observing independent emergence and disappearance of individual response peaks between sessions (Fig. 1f, Supplementary Fig. 3). In some cases, this occurred suddenly or randomly between sessions, but in many cases, such changes occurred gradually and continuously. Note that the unstable responses to MOV were not found only in neurons responding selectively to the MOV stimulus, but also in the subset of neurons responsive to both stimuli, such as the neuron in Fig. 1f, in which representational drift in MOV responses

could be directly contrasted with highly stable responses to the PDG stimulus.

In contrast to measures of reliability occurring across trials within a recording session, here we define "stability" as the consistency of the average neuronal responses to the same visual stimulus over many sessions. To quantify changes in single-neuron response stability, we defined the "representational drift index" (RDI) between two sessions as the difference between the within-session between-trial correlation coefficient ($CC_{WS}$) and the between-session between-trial correlation coefficient ($CC_{BS}$), normalized by the sum of the two (Fig. 1h). Here, the $CC_{WS}$ provides a baseline measurement of the robustness of a neuron's response and the $CC_{BS}$ provides a measurement of the robustness of the signal across sessions. Note that the index will tend toward 0 for a neuron that is completely stable across sessions and tend toward 1 for a neuron that is robust within a session, but not across sessions. Such a correlation-based metric is necessary to account for individual neurons exhibiting multiple response peaks of varying degrees of amplitude during the time course of the movie. Pooling data from all dual-responsive neurons across all imaging fields ($n = 824, 808, 793, 830, 761, 698$ neurons from 13, 12, 12, 13, 11, 9 imaging fields for sessions 2–7 respectively), we found that RDI values for MOV responses were on average significantly greater than those for PDG on all sessions ($p < 0.001$ for sessions 2–6, $p < 0.01$ for session 7; linear mixed-effects model, fixed effect for stimulus, random effect for mouse), and that the representational drift was not random, but progressively increased over sessions for most mice (Fig. 1i, Supplementary Fig. 5a–c; PDG vs. MOV average RDI curve slopes, $p < 0.05$, two-tailed paired-sample t-test). The RDI values were not significantly different when including neurons responsive to single stimuli, when compared to only dual-responsive neurons (comparing single stimulus responsive to dual-responsive, $p > 0.05$ for all weeks for both stimuli; linear mixed-effects model).

Despite a robust difference in RDI between PDG and MOV stimuli on average, we observed some variability between mice. While most clearly exhibited a difference in stability between the stimuli, this difference was either small or not significant in a subset of mice (Supplementary Fig. 5a, mice 5, 11, 12). In addition, while some mice exhibited a progressive divergence between the PDG and MOV RDI curves (mice 1, 3, 4, 6, 8, 9), others exhibited higher baseline RDI levels for MOV stimuli (mice 2, 7, 10). The reasons for these differences were unclear given that there were no identifiable correlated differences between groups of mice. For instance, we found no significant relationship between animal age at the first imaging session and the final RDI values for either stimulus (Supplementary Fig. 5d). Furthermore, the exclusion of neurons via thorough a priori manual inspection eliminated any artifacts that may have resulted from misalignment of neurons across sessions in some mice and repeating our analysis using different ROI quality thresholds yielded robust results (Supplementary Fig. 1). One potential source of variability between mice is that craniotomies for cranial window implants can cause inflammation and microglial activation[57], which may, in turn, result in increased synaptic turnover in superficial layers[58]. It is possible that exposure of the brain in the cranial window preparation may be partially responsible for the differences between mice we observe. Thin-skull preparations for transcranial 2-photon imaging have been shown to result in a relatively low degree of induced synaptic plasticity[15]. To test whether the surgical preparation contributed to the drift in MOV responses, we performed the same chronic imaging experiments in mice ($n = 4$) with a thin-skull preparation and found a similar level of differential stability between the two stimuli (Supplementary Fig. 6a, b; $p < 0.001$ for sessions 2, 5, 6, $p < 0.01$ for session 7, $p < 0.05$ for session 4, linear mixed-effects model).

**Differences in RDI between stimuli are not explained by stimulus temporal structure**. Another potential concern is the difference in presentation structure between the two stimuli. Does the continuous nature of the MOV stimulus, compared to the more discrete presentation of gratings (presentations separated by inter-stimulus intervals), lead to the resulting differences in RDI? To explore this, we repeated the experiments in another group of mice ($n = 4$) using stimuli with matched temporal structures. In addition to the original PDG and MOV stimuli, we presented: (1) the PDG stimulus with concatenated grating presentations and no interstimulus gray screens (PDG continuous), meant to match the temporal structure of the original MOV stimulus, and (2) the MOV stimulus split into 12 presentation periods of 2 s each, broken up by 4 s interstimulus gray screens (MOV discrete), meant to match the structure of the original PDG stimulus (Supplementary Fig. 7a, b). As in our previous results, RDI curves were significantly different between PDG and MOV (Supplementary Fig. 7c, $p < 0.01$ for session 2, $p < 0.001$ for sessions 3–7). Significant differences were also found comparing both sets of temporal-structure-matched stimuli: RDI values for MOV were greater than those for PDG continuous ($p < 0.001$ for sessions 3, 5, 7, $p < 0.01$ for session 4, $p < 0.05$ for session 6, linear mixed-effects model), and RDI values for MOV discrete were greater than those for PDG ($p < 0.01$ for session 2, $p < 0.001$ for sessions 3–7; linear mixed-effects model). In addition, the two new stimuli exhibited a similar degree of differential drift as was seen from the original stimuli (Supplementary Fig. 7d, $p < 0.05$ for session 2, $p < 0.001$ for sessions 3–7, linear-mixed effects model). Since neither stimulus manipulation was sufficient to eliminate differences between the stimuli, these results indicate that the greater degree of representational drift to MOV stimuli is not attributable to the temporal structure of the stimulus.

Finally, in the context of perceptual learning, it has been demonstrated that repeated presentation of grating stimuli can induce gain changes in cortical responses over many days during the performance of visual perception tasks[37,59]. Passive stimulus viewing alone may be sufficient to induce perceptual learning and incite changes for both gratings and natural stimuli[37,60,61], possibly with differences across stimuli. To address this, we ran additional experiments in which mice ($n = 3$) were shown the PDG and MOV stimuli on Day 0, and then again on Day 42 without either of the visual stimuli presented in the interim. These experiments also produced representational drift (Supplementary Fig. 8a), as well as differential RDI values between the two stimuli as was seen in the original experiments (Supplementary Fig. 8b, $p < 0.001$, linear mixed-effects model). As a result, perceptual learning is not sufficient to account for the observed differences in representational drift.

**Responses to natural movies exhibit greater drift independent of the magnitude**. One factor that could explain differential stability on the level of single neurons is the wide diversity of stimulus responsiveness that exists in sensory cortices[38,39]. A recent study in mouse V1 found considerable differences in long-term stability between strongly and weakly visually responsive neurons[40]. In addition, it is possible that early accounts of stable cortical neurons from electrophysiology data may be influenced by the biased sampling of, particularly active neurons. We asked if a similar relationship between visual responsiveness and stability could be found in our data. On average, the neurons responsive to both stimuli with the strongest session-average z-scored responses were more stable than those with the weakest (Fig. 2a, Supplementary Fig. 9a). The difference was modest but statistically significant for both stimuli (0.04 difference in median RDI between the top and bottom quartiles for PDG, $p < 0.01$; 0.07 difference for MOV, $p < 0.001$; Wilcoxon rank-sum test). Lack of striking, conclusive results from this analysis led us to ask if there may be further differences related to responsiveness that are not observable on the single-neuron level.

The sparse and episodic structure of V1 neuron responses to naturalistic stimuli has been well-characterized[62–64] and was qualitatively visible in our response data (Supplementary Fig. 3). Most, but not all, neurons responsive to MOV exhibited more than one time-distinct response "event" across the course of the 30 s stimulus (Figs. 1f and 2b, Supplementary Fig. 3). We asked if a relationship between response strength and stability was present at the level of the individual visual response events within each neuron. For both stimuli, we used $\Delta F/F$ data to define response event periods based on the statistical significance of each frame's single-trial responses compared to single-trial average baseline responses, performed separately for each session (see "Methods"). PDG and MOV responses in dual-responsive neurons yielded, on average, ~1.9 and ~2.4 events per neuron, respectively, with a sizable subset of neurons responding to MOV with 3 or more events (51.6% of dual-responsive neurons, 57.2% of MOV responsive neurons, Fig. 2c). The event rate distribution for PDG was skewed further towards 1–2 events per neuron, reflecting selectivity towards only 1 or 2 grating orientations and occasional weak responses to neighboring orientations (Fig. 2c). Events were categorized as either growing, decaying, or remaining static over time-based on a statistical comparison of their single-trial z-score values on the first two sessions versus the last two sessions. We found that on average across all imaging fields, PDG responses exhibited a much higher proportion of events that remained static between sessions than MOV responses did (~73% vs. ~54% for PDG and MOV respectively, $p < 0.001$, two-tailed paired-sample t-test). Of the remaining 46%

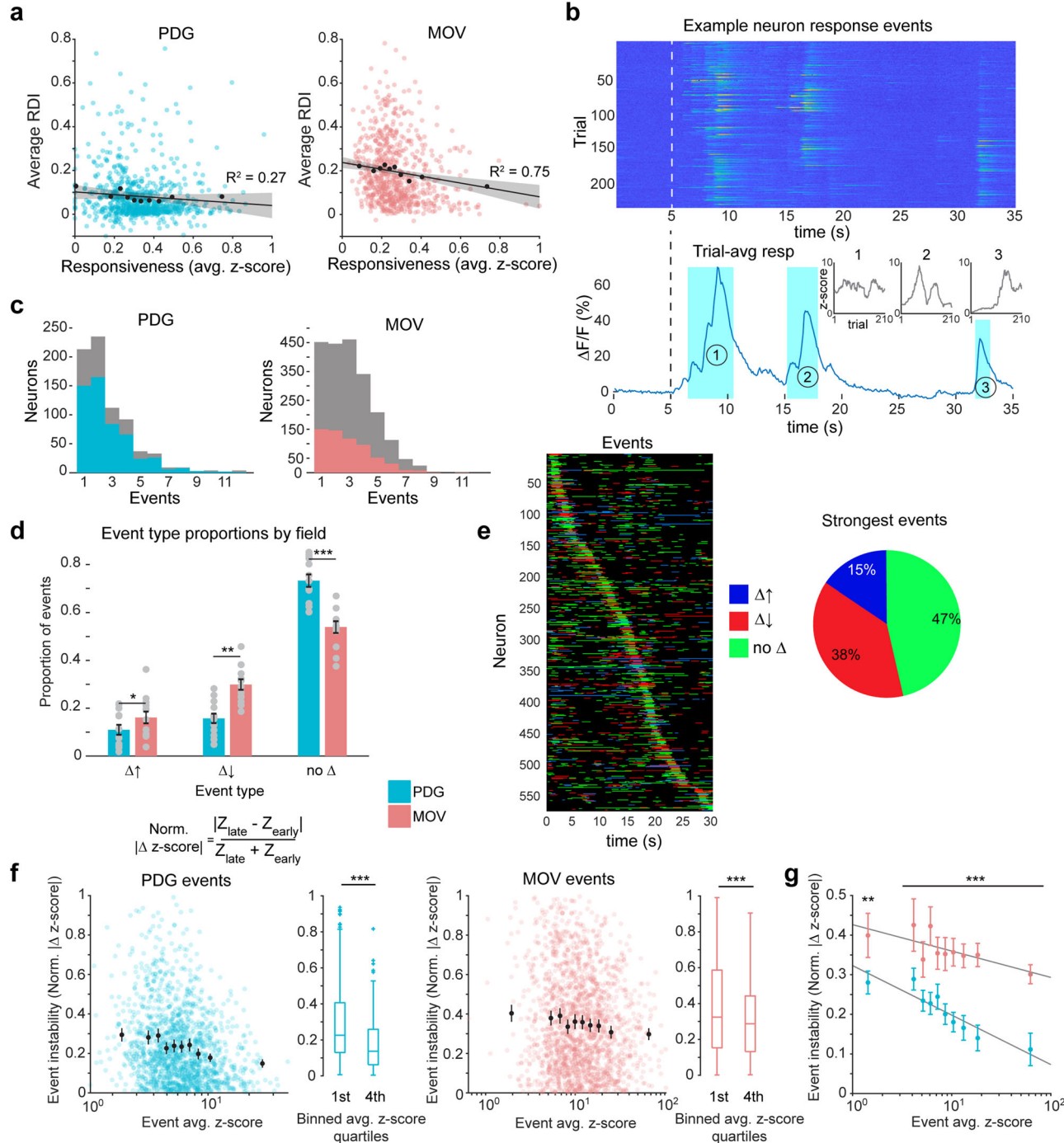

of MOV events, ~65% were events that decayed and ~35% were those that grew between sessions (Fig. 2d). Examining the strongest event periods (events surrounding neurons' peak response times) revealed that while a plurality of these events remained static over time, over half were classified as either growing or decaying (Fig. 2e).

To compare an event's magnitude and its stability more directly, we defined an event's instability based on the normalized difference between its average "late" z-score and its average "early" z-score (last two versus first two sessions, respectively, see "Methods"). This revealed that for both stimuli, an event's stability increased as a function of its overall response strength (Fig. 2f, Supplementary Fig. 9a). This trend occurred to a similar extent for both stimuli, and event instability was greater on

average for MOV events than they were for PDG events regardless of event magnitude (Fig. 2g). As a result, differences in stability between the two stimuli cannot be explained by the presence of higher magnitude events in PDG responses compared to MOV responses. In addition, we wondered if we could link an event's stability to its redundancy across all neurons in its respective population, as previous findings show stability in sparse population activity[12]. Indeed, event stability seemed to decrease as a function of redundancy across all other neurons in the population (Supplementary Fig. 9b, comparing first and last quartiles, $p < 0.001$ only for PDG, Wilcoxon rank-sum test). In line with this, we found that event redundancy decreased significantly as a function of event response magnitude for both stimuli (Supplementary Fig. 9c, comparing first and last quartiles, $p < 0.001$ for

**Fig. 2 Characterization of dynamic response events underlying single-cell responses. a** Single neuron RDI as a function of responsiveness (session-average z-score of ΔF/F activity). Each colored dot is one neuron; black dots are 10th percentile binned means; black line is a linear fit of the binned data; shaded area indicates 95th percent confidence interval of the linear fit. Neurons are z-scored using the entire recording on a given session. Data are shown for all neurons responsive to both PDG and MOV ($n = 736$ neurons). **b** Response events from one example neuron. Top: ΔF/F responses, all trials across all recording sessions. Bottom: Trial-averaged response. Shaded areas indicate identified events. Insets: z-score trajectories (smoothed using 30-point moving average) across all trials for the three events in the example neuron. **c** Number of response events per neuron. Left: all PDG responsive neurons are shown in gray, dual-responsive neurons shown in color. Right: all MOV responsive neurons shown in gray, dual-responsive neurons shown in color. **d** Proportions of event types (growing, decaying, and static) in responses to both stimuli. Event type is determined by z-scoring an event waveform's single-trial responses and comparing the distributions of these values between the first two sessions (60 trials) and the last two sessions (60 trials; Wilcoxon rank-sum test). Gray dots are individual fields. Bar data shown are mean proportions across all imaging fields ± sem ($n = 13$ imaging fields; growing events $t_{12} = -2.6$, $p = 0.04$; decaying events $t_{12} = -4.2$, $p = 0.001$; static events $t_{12} = 4.8$, $p = 3.8 \times 10^{-4}$; two-tailed paired-samples t-test; *$p < 0.05$, **$p < 0.01$, ***$p < 0.001$). **e** Visualization of MOV response event magnitude changes. Left: bands indicate event periods for each neuron, colored by event type. Neurons are ordered by time of maximum trial-averaged response. Right: proportions of event types for neurons' peak responses (diagonal of the left plot). Data are shown for all dual-responsive neurons. **f** Event instability (normalized delta z-score) as a function of event magnitude (session-average event z-score). Each colored dot is one event; black dots are 10th percentile binned means ± 95th percent confidence interval. Box plots are the first quartile of data tested against the fourth quartile ($n = 365$ events per quartile for PDG, $Z = 7.3$; $n = 441$ events per quartile for MOV, $Z = 3.4$; ***$p < 0.001$, two-sided Wilcoxon ranksum test). Data are shown for all dual-responsive neurons. Boxplots are centered on the median, boxes extend to first and third quartiles, whiskers extend to 1.5 times the interquartile range or minima/maxima in the absence of outliers. **g** MOV events are less stable than PDG events independent of event magnitude. Binned means (10th percentiles) using data from (**f**) for both stimuli shown together. Data from all events across both stimuli were pooled to determine bin edges, events from each stimulus were then binned separately. Error bars represent the 95th percent confidence interval, gray lines are linear fits of the data. Significance markers indicate comparison of PDG and MOV values in each bin (**$p < 0.01$ for first bin, ***$p < 0.001$ for all other bins; two-sided Wilcoxon rank-sum test).

both stimuli, Wilcoxon rank-sum test). This contravenes the possibility that stronger events may be driven by visual features of the movie with higher bottom-up salience and thus may be represented by a greater fraction of neurons in the population.

**Differential stability is not explained by the state of arousal.** We performed a number of control experiments to account for external factors that could contribute to these results. A previous study reported that despite stable orientation tuning across days, pupil size was positively correlated with trial-to-trial variability[36]. We used an infrared camera to track the pupils of a subset of mice ($n = 4$) to determine whether the differential stability could be related to differences in a mouse's arousal state across sessions as measured by pupil movement or pupil size[65]. We observed that large eye movements were rare and most movements consisted of brief deviations from an otherwise stable average position that changed minimally between sessions (Supplementary Fig. 10a, b). Pooling data across mice, changes in the amount of eye movement relative to the reference session were similar between stimuli (Supplementary Fig. 10c; $p > 0.05$, two-tailed paired-sample t-test). Comparing changes in eye movement over time to RDI curves for individual mice revealed no clear relationship between eye movement and stability for either stimulus (Supplementary Fig. 10d), although pooled data showed that eye movements may decrease over time slightly more for MOV (Supplementary Fig. 10e). Changes in pupil size over time showed similar results. First, we tested the effect of pupil size on visual response magnitude, as has been demonstrated previously[65,66]. We compared each trial's average pupil size (relative to session-average pupil size) to its response gain (Methods) and found a moderate correlation for both stimuli (Supplementary Fig. 11a, b; MOV Pearson $r = 0.31$, $p < 0.001$, PDG Pearson $r = 0.41$, $p < 0.001$). Changes in pupil size (relative to reference session) were not significantly different between stimuli (Supplementary Fig. 11c; $p > 0.05$, two-tailed paired-sample t-test). For both individual mice and averages across mice, we found no clear differences in pupil change over time (Supplementary Fig. 11d, e), and individual mice showed no clear relationship between change in pupil size and representational drift over time (Supplementary Fig. 11e). In summary, though there were minor changes in pupil area across sessions, the corresponding fluctuations in mouse

arousal state are not sufficient to explain progressive increases in RDI, nor the difference in RDI between MOV and PDG stimuli.

Next, we considered the possibility that any subtle shifts in a neuron's spatial receptive field across sessions would manifest only in MOV responses and not PDG responses. While changes in a neuron's receptive field would likely capture different visual information for the MOV stimulus, this might not be the case for PDG due to its repeated spatial pattern. To account for this, we showed a subset of mice ($n = 4$) a spatial receptive field mapping stimulus (presented last on every session) to measure neurons' preferred altitude and azimuth on a given session ("Methods"). In all tested mice, changes in neurons' preferred altitude and azimuth were minimal (Supplementary Fig. 12a) and showed no correlation with higher RDI (Supplementary Fig. 12b, c), indicating that receptive field shifts did not influence our findings.

**Representational drift is distributed across cortical layers and cell types.** Because the distinct layers of the visual cortex have stereotyped inputs, outputs, and connectivity within and between other layers[67], we wondered if they might show differences in their capacities for representational drift. As L4 is the primary recipient of input from the lateral geniculate nucleus (LGN), we hypothesized that it would be comprised of neurons with relatively stable responses to the MOV stimulus compared to neurons in L2/3 and L5, which receive more processed input[68]. However, a recent study of natural movie responses in the mouse visual cortex found higher stability of firing rate and stimulus tuning in L2/3 and L5 neurons compared to L4 neurons over multiple days[49]. To investigate differences between layers, we chronically implanted a subset of mice ($n = 4$) with a custom glass micro prism that granted optical access to nearly the full cortical column[69] (Fig. 3a, b), and allowed us to separate neurons by cortical layer based on ROI density (Fig. 3c). Comparing average RDI values within each layer revealed that the cortical layer did not have a significant effect on the instability of individual neurons (Fig. 3d, e; stimulus type $p = 3.32 \times 10^{-11}$, layer $p = 0.12$, two-way ANOVA). This homogenous distribution of neurons exhibiting representational drift suggests that the transformation of presumably stable visual input to unstable neural responses occurs either at or before the first stage of information processing in the cortex and propagates as the signal continues.

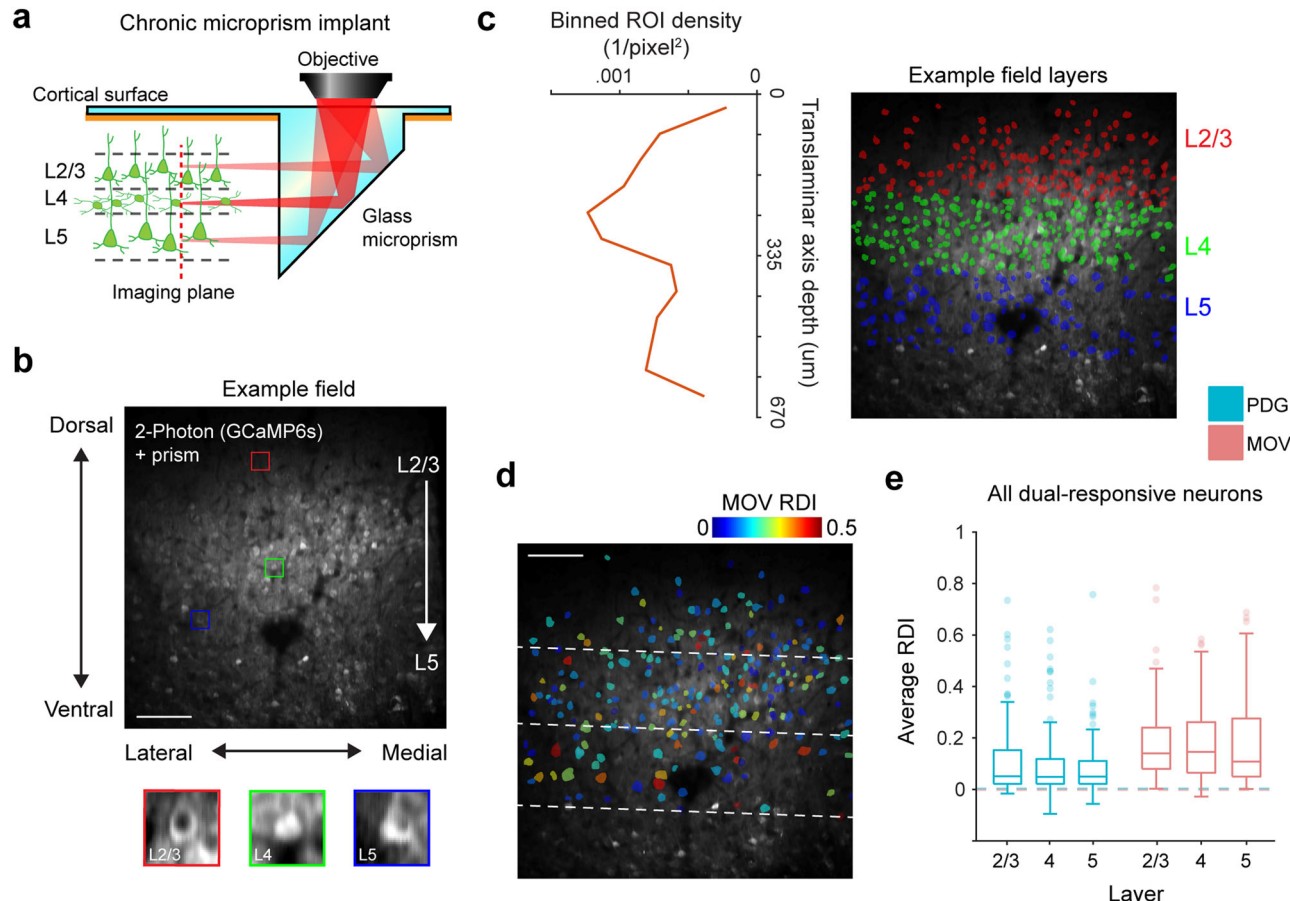

**Fig. 3 Translaminar imaging shows equal RDI distributions across layers. a** Schematic of glass microprism placement in V1. The red dotted line depicts the translaminar imaging plane, which is rotated 90° from the original horizontal plane and spans ~700 μm of the cortical column, capturing neurons in L2-5 for a typical recording. **b** Example average fluorescence image from a chronic imaging session of a prism field. Colored boxes are zoomed-in example cells (red: L2/3 pyramidal neuron, green: L4 stellate neuron, blue: L5 pyramidal neuron). Scale bar = 100μm. **c** Delineation of cortical layers. Left: ROI density of binned pixel windows perpendicular to the cortical column axis (corresponds to right subfigure); layers are determined by finding peak density and assigning a 140 μm window around it as L4, and then a further 150 μm from the L4 deep boundary as L5. Right: example field is shown in (**b**) with an overlay of all ROIs colored by layer; dotted line is the translaminar axis. **d** Example field shown in (**b, c**) with an overlay of ROIs of all well-tracked neurons responsive to MOV, colored according to MOV RDI. Dotted lines are layer boundaries. **e** Session-averaged RDI distributions by stimulus and layer, using all dual-responsive neurons recorded in prism fields ($n = 64$ L2/3 neurons, 111 L4 neurons, 121 L5 neurons from 4 mice). No significant difference was found between layers, a significant difference was found between stimuli (layer $F_{2,588} = 2.06$, $p = 0.12$; stimulus $F_{1,588} = 45.7$, $p = 3.32 \times 10^{-11}$; two-way ANOVA). Dotted lines indicate control RDI (as in Fig. 1h, i) using all dual-responsive neurons. Boxplots are centered on the median, boxes extend to first and third quartiles, whiskers extend to 1.5 times the interquartile range or minima/maxima in the absence of outliers.

We next investigated if MOV representational drift varied by cell type. Inhibitory interneurons exhibit broader and less selective tuning than excitatory cells, believed to be due to promiscuous sampling from local excitatory populations[70], which may result in comparatively stable visual responses. Inhibitory interneurons in mouse motor cortex[71] and zebra finch HVC[72] (but see ref. [73]) have been shown to exhibit considerable long-term stability alongside less stable single excitatory neurons in the context of a motor task. To investigate this, we repeated our experiments in L2/3 of transgenic mice expressing GCaMP6s in GAD2 + inhibitory neurons ($n = 4$, Fig. 4a, b). Inhibitory neurons exhibited clear visually-evoked responses, and as expected most neurons responsive to PDG were relatively broadly tuned (Fig. 4c). As in our excitatory population results, of all 232 well-tracked neurons, only a subset ($14 \pm 5\%$) were visually responsive to both PDG and MOV (Fig. 4d). We considered this subset of dual-responsive neurons as we did for the excitatory populations. Similar to the between-session changes observed in excitatory neurons, we observed the appearance and disappearance of individual visual response events within neurons across sessions (Fig. 4c). As for excitatory neurons, we found that

responses to MOV exhibited greater representational drift than responses to PDG (Fig. 4e, f; $p < 0.05$ for sessions 4, 5, $p < 0.01$ for session 6, $p < 0.001$ for session 7, linear mixed-effects model). However, we found that the degree of drift may not be consistent across inhibitory cell types, as inhibitory neurons sharply tuned for orientation exhibited higher RDI than those broadly tuned for orientation (Supplementary Fig. 13). These results indicate that progressive drift of responses to naturalistic stimuli is present across the entire cortical network.

**Representational drift is accompanied by changes in population correlation structure.** To further investigate the underpinnings of the stimulus-dependent instability, we hypothesized that the greater representational drift in the MOV responses might be due to the higher-order image statistics of the stimulus. We reasoned that the additional cortical processing required for encoding more complex visual features might lead to more variable representation compared to that of oriented gratings. To test this, we performed parametric phase scrambling on the

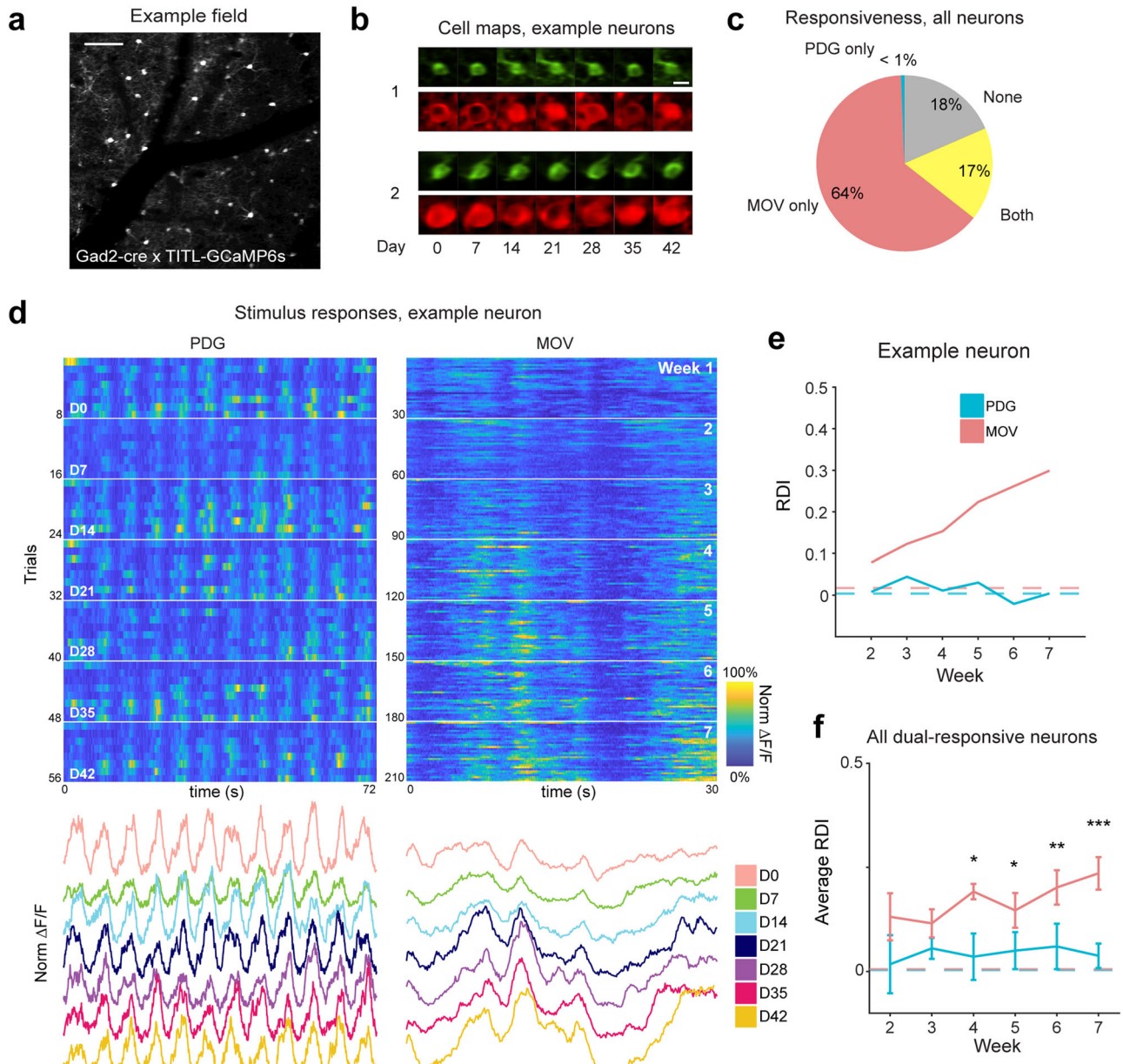

**Fig. 4 Inhibitory neuron populations also exhibit representational drift to MOV stimuli. a** Average fluorescence image from an example field of inhibitory neurons in L2/3 of V1 of GAD2-Cre × TITL2-G6s mice. Scale bar = 100 μm. **b** Example images of well-tracked neurons (see "Methods") from the field in (**a**) on all recording days, green and red colors are the same as in Fig. 1c. Scale bar = 15 μm. **c** Fluorescence traces ($\Delta F/F$) for one example neuron. Trials are concatenated across sessions. Left: responses to PDG. Right: responses to MOV. White horizontal lines indicate divisions between recording sessions. Heatmaps for each stimulus are co-normalized. Below each heatmap are trial-averaged responses colored by session. **d** Average percentage of neurons responsive to each stimulus (MOV only: 59.4 ± 1.9% sem, PDG only: 0.6 ± 0.6%, both: 14.4 ± 5.7%, none: 25.6 ± 6.3%). **e** RDI curves for the example neuron shown in (**c**); dotted line indicates control RDI for this cell (see "Methods"). **f** Average RDI curves from neurons across all imaging fields; error bars are ± sem; data shown for dual-responsive neurons present on any given session ($n = 33, 34, 34, 33, 33, 31$ neurons from 4 fields for sessions 2–7 respectively); significance markers indicate the comparison of each session's PDG RDI values and MOV RDI values ($F_{1,64} = 0.6$, $p = 0.44$, $F_{1,66} = 0.3$, $p = 0.61$, $F_{1,66} = 6.6$, $p = 0.01$, $F_{1,64} = 5.6$, $p = 0.02$, $F_{1,64} = 7.9$, $p = 0.007$, $F_{1,60} = 15.1$, $p = 2.5 \times 10^{-4}$ for sessions 2–7 respectively; two-tailed $F$-test using a linear mixed-effects model, fixed effect for stimulus, random effect for mouse; *$p < 0.05$, **$p < 0.01$, ***$p < 0.001$). Dotted lines indicate control RDI as in previous figures.

original MOV stimulus[74], which randomized the phase structure of the images in the movie while maintaining the amplitude spectrum and lower-level image properties. To a subset of mice ($n = 2$), we presented the PDG stimulus in addition to a stimulus consisting of 3 versions of the MOV stimulus: 100% phase scrambled, 50% phase scrambled, and 0% phase scrambled (the original movie), presented for 20 repeats each in random trial order (Fig. 5a). However, we observed no major differences between the RDI curves for the three MOV stimuli (Fig. 5b), indicating that the increased representational drift is not due purely to higher-order image statistics.

Our earlier results indicate that what separates stable and unstable visual responses in V1 is the identity of the visual input, and it would be inaccurate to conclude that a neuron itself is

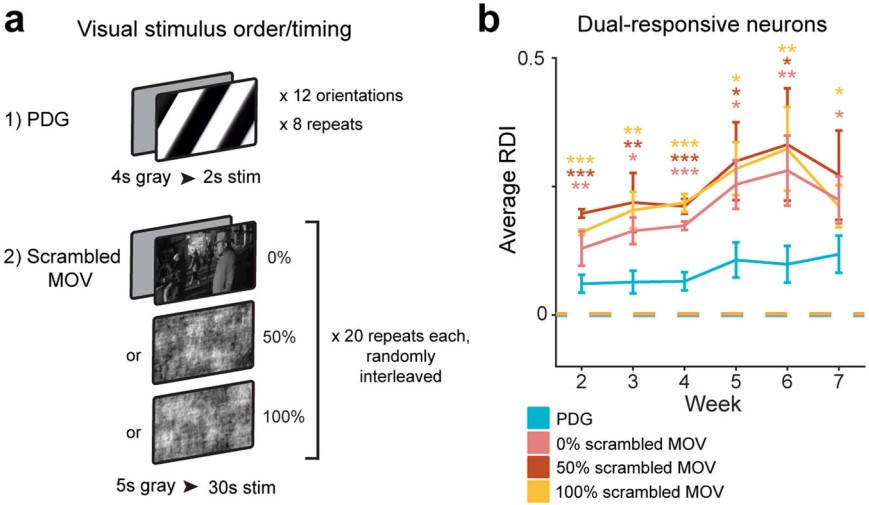

**Fig. 5 Stability is not dependent on the higher-order statistics of the visual stimulus. a** Visual stimuli, where the original 30 repeats of MOV is replaced with 20 repeats each of 0%, 50, and 100% phase-scrambled versions of the original movie, randomly interleaved. Bottom screens depict the same freeze-frame from each of the movie versions. **b** Average RDI curves from all imaging fields; error bars are ± sem; values are calculated using only neurons that are present on any given session ($n = 120, 118, 118, 119, 113, 111$ neurons from 3 fields for sessions 2–7 respectively). Average PDG RDI is significantly different from average MOV RDI for all three movie versions (0% scramble $F_{1,380} = 10.7$, $p = 0.001$, $F_{1,368} = 6.6$, $p = 0.01$, $F_{1,378} = 40.2$, $p = 6.6 \times 10^{-10}$, $F_{1,374} = 4.9$, $p = 0.03$, $F_{1,362} = 8.1$, $p = 0.005$, $F_{1,344} = 4.9$, $p = 0.03$ for sessions 2–7 respectively; 50% scramble $F_{1,380} = 52.7$, $p = 2.2 \times 10^{-12}$, $F_{1,368} = 7.4$, $p = 0.007$, $F_{1,378} = 24.7$, $p = 1.0 \times 10^{-6}$, $F_{1,374} = 4.2$, $p = 0.04$, $F_{1,362} = 5.3$, $p = 0.02$, $F_{1,344} = 2.9$, $p = 0.09$ for sessions 2–7 respectively; 100% scramble $F_{1,380} = 26.8$, $p = 3.6 \times 10^{-7}$, $F_{1,368} = 9.4$, $p = 0.002$, $F_{1,378} = 27.9$, $p = 2.1 \times 10^{-7}$, $F_{1,374} = 5.9$, $p = 0.02$, $F_{1,362} = 9.3$, $p = 0.002$, $F_{1,344} = 6.5$, $p = 0.01$ for sessions 2–7 respectively; two-tailed $F$-test using a linear mixed-effects model, fixed effect for stimulus, random effect for mouse; *$p < 0.05$, **$p < 0.01$, ***$p < 0.001$; color of asterisk corresponds to MOV stimulus version compared to PDG). Dotted lines indicate control RDI, as in previous figures.

intrinsically stable or unstable. This raises the possibility that the stability in different conditions is a function of the particular functional inputs. We investigated whether or not the observed stimulus-dependent representational drift in single neurons could be linked to stimulus-dependent changes in the correlation structure of the local population, as even in the primary sensory cortex neuronal responses are strongly influenced by local circuit activity[34,75–77]. Subnetworks of neurons co-responsive to natur-alistic stimulus features may have less pre-existing connectivity compared to subnetworks co-tuned to specific orientations of the PDG stimulus, which have been found to exhibit high interconnectivity[47,48,78]. Does the MOV population correlation structure change across weeks while the PDG correlation structure remains consistent? To answer this, on every session for both stimuli we calculated pairwise between-neuron signal correlations (SC), which capture shared responses to stimulus-driven input, and noise correlations (NC), which capture shared trial-to-trial variability (see "Methods"). Visualizing SC structure on the final recording session ($D_{final}$) versus the reference session ($D_0$) for individual mice showed larger differences for MOV than for PDG (Fig. 6a), and absolute changes in mean SC across neurons were greater on average for MOV than for PDG in 12 out of 13 fields (Fig. 6b, c). Consistent with another study investigating cross-session changes in grating stimulus correlation structure[35], both the MOV and PDG SC structures became gradually less similar to $D_0$ over time, as measured by the Pearson correlation between each matrix and the $D_0$ matrix (Fig. 6d). However, the MOV correlation structure exhibited greater divergence as a function of time since $D_0$ (Fig. 6d, comparing curves between stimuli, $p < 0.05$ for session 2, $p < 0.01$ for session 6, $p < 0.001$ for sessions 3, 4, 5, 7, two-tailed paired-sample t-test). Note that changing SC structure does not necessarily result from single-cell representational drift, as large, coordinated shifts in activity among groups of neurons (e.g., arousal-related gain changes) would result in representational drift without influen-cing SCs among those neurons. Repeating these analyses for noise

correlations revealed a weaker effect than that of SC, though the trend was in the same direction (Supplementary Fig. 14), possibly due to relatively low levels of noise correlations in our data. Previous studies indicate that NC measurements can vary significantly and may be affected by a variety of factors, such as spike rates and behavioral state[75,79]. For this reason, we cannot conclusively determine whether drift is present in NCs as well. In summary, the differential stability of responses to PDG and MOV stimuli does not appear to be related to higher-order statistical features in the MOV stimulus, but rather to the decreased stability of the co-tuned neuronal ensembles driven by each stimulus.

## Discussion

Here we investigated the extent to which representational drift exists in populations of individual neurons through a comparison of neural responses to artificial (PDG) and naturalistic (MOV) visual stimuli in the primary visual cortex of awake mice over many weeks. We found that neurons exhibited highly consistent orientation tuned responses to the grating stimulus overall recording sessions (Fig. 1, Supplementary Fig. 2), consistent with the previous work[18,22,36,40]. However, many of the same neurons displayed less stable representations of the natural movie stimulus (Figs. 1–2, Supplementary Figs. 3–5). Contrary to our hypothesis that geniculate inputs to L4 may grant it more consistent stimulus representation, translaminar imaging revealed that this differential stability existed not only in L2/3 and L5 but also in L4 (Fig. 3). Repeating our experiments in populations of inhibitory interneurons yielded a similar difference in stability between the two stimuli (Fig. 4). Importantly, the observed drift could not be explained by external factors such as eye movement, arousal, or surgical preparation (Supplementary Figs. 6, 10, and 11). In addition, the higher-order image statistics of natural sti-muli were not found to contribute to the differential drift between stimuli (Fig. 5). Finally, we found that the drift observed in MOV responses was accompanied by changes in the population

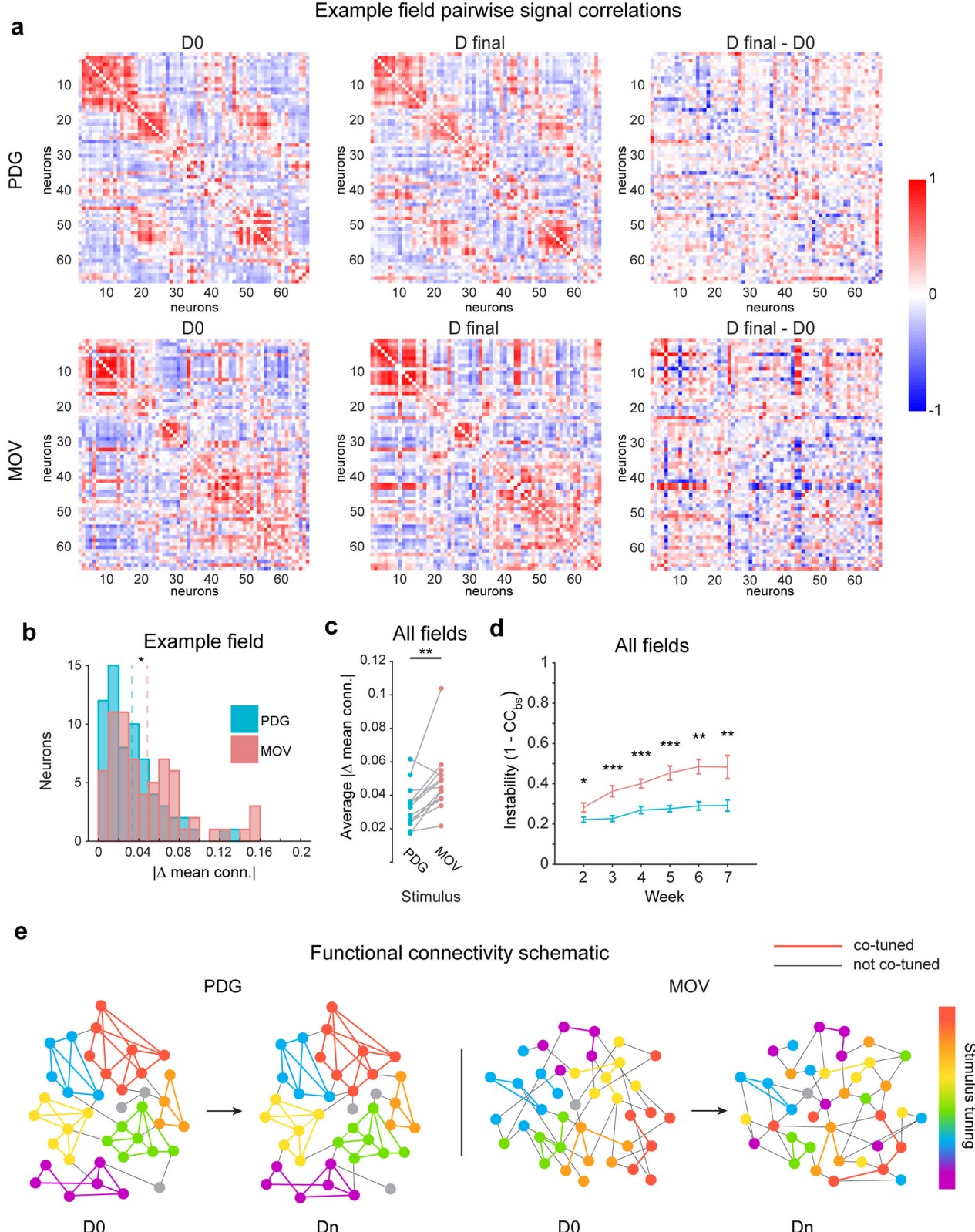

**a** Example field pairwise signal correlations

**b** Example field

**c** All fields

**d** All fields

**e** Functional connectivity schematic

correlation structure, potentially due to a lack of strongly connected iso-tuned subnetworks that respond to grating stimuli (Fig. 6).

From work that utilizes artificial stimuli such as gratings, compelling evidence suggests that basic neuronal response properties such as orientation tuning, spatial frequency tuning, and size tuning are remarkably stable over time[18,22,36]. However,

another recent study reports that naturalistic stimulus responses in the mouse visual cortex exhibit significant representational drift[49]. By performing a direct comparison of artificial and naturalistic visual stimulus-response in the same animals, our experiments allowed us to reconcile this discrepancy by showing that response stability is dependent on stimulus, and not simply an intrinsic property of a given neuron.

**Fig. 6 Between-neuron signal correlation stability is stimulus-dependent. a** Pairwise signal correlations on session 1 (left), final session (middle), and their difference (right) for one example field. Neurons are sorted by the time of peak response on $D_0$ for each stimulus. Data are shown for all neurons responsive to both stimuli. **b** Distributions of the single-neuron average change in signal correlations between first and final sessions for example field in A. Dotted lines are means for each stimulus ($n = 66$ neurons, $Z = 2.3$, $p = 0.02$, two-sided Wilcoxon rank-sum test; *$p < 0.05$). **c** Field-average changes in signal correlation between first and final sessions. Data shown for all fields. ($n = 13$ imaging fields; $t_{12} = 3.8$, $p = 0.003$, two-tailed paired-sample $t$-test; **$p < 0.01$). **d** Average instability of signal correlation matrices with respect to the first session over time ($1 - CC_{BS}$; where $CC_{BS}$ is the 2D cross-correlation between signal correlation matrices). Data shown for all fields. Error bars are ± sem; significance markers indicate comparison of PDG and MOV values on the given session ($n = 13$, 12, 12, 13, 11, 9 imaging fields for sessions 2–7 respectively; $t_{12} = 2.2$, $p = 0.04$, $t_{11} = 5.8$, $p = 1.2 \times 10^{-4}$, $t_{11} = 6.9$, $p = 2.4 \times 10^{-5}$, $t_{12} = 5.0$, $p = 3.1 \times 10^{-4}$, $t_{10} = 4.5$, $p = 0.001$, $t_8 = 3.4$, $p = 0.009$, for sessions 2–7 respectively, two-tailed paired-sample $t$-test; *$p < 0.05$, **$p < 0.01$, ***$p < 0.001$). **e** Schematic depicting the relationship between stimulus tuning stability and shifts in functional connectivity over time.

Further characterization of the representational drift observed in MOV responses revealed two important results. First, we showed that differential stability between stimuli existed not only at the single-neuron level but also at the scale of individual response events, wherein response events during the MOV stimulus were more likely to experience bidirectional amplitude changes across sessions than those that occurred during the PDG stimulus (Fig. 2d, e). Second, consistent with previous work demonstrating a relationship between response strength and stability[40], we found such a correlation to be evident on the level of individual events (Fig. 2f, g, Supplementary Fig. 9a). These results indicate representational drift may operate at the level of specific functional inputs, rather than at the level of single neurons. Deitch et al. found similar single neuron representational drift that appeared to be dominated by changes in firing rate rather than changes in stimulus tuning[49]. However, our results indicate that these two properties are interconnected in that changes in firing rate can manifest as changes in tuning over long time scales. We also noted an inverse relationship between event response strength and event redundancy, wherein the strongest events are not only the most stable but also the least redundant across other neurons in the recorded population (Supplementary Fig. 9b, c). Neurons that primarily respond with strong, stable events that are relatively sparsely represented across the population may constitute a response class previously characterized as "soloists", and those that primarily respond with weak, redundant events may represent "choristers" more strongly coupled to the local population[80,81]. This is consistent with the developing characterization of sensory cortices as being dominated by subnetworks of particularly stable and robustly responding neurons against a background of weakly responsive neurons[25,38,39,79].

One question still outstanding is the extent to which representational drift is present throughout the visual hierarchy. Presumably, repeated visual input is encoded in a consistent fashion within the retina and the earliest stages of the visual pathway. Based on the results that unstable responses arise as early as layer 4 of V1, one hypothesis might be that the primary visual cortex as a whole represents a relatively intermediate processing stage in which single neurons are permitted greater coding flexibility and that LGN, which receives input directly from the retina, exhibits greater stability. However, given recent results showing that retinogeniculate and retinocollicular boutons are modulated by arousal[82,83], representational drift might begin even earlier. Future studies using glass microprisms or GRIN lenses to allow chronic optical access to LGN[83] will allow elucidation of these possibilities. Finally, it remains to be seen whether stimulus representations stabilize as signals propagate to higher visual areas, or if the capacity for drift is preserved across the entire visual pathway. To answer this, another future direction would be to perform similar longitudinal imaging in higher visual areas concurrently with recordings from V1 in the same animals.

Representational drift appears in varying degrees across different brain structures, and most prevalently outside of the sensory cortex. In motor and parietal cortices, evidence suggests that population-level activity patterns remain intact, even as the underlying neurons representing task or motor information experience significant drift[7,8,19,20,72], although see[4,6]. Several studies have identified strong representational drift in hippocampal place cells that exhibit dynamic cognitive maps of the environment[27,28,84,85], though results vary across hippocampal subregions[86]. One possible interpretation is that the opportunity for representational drift increases with distance from the sensory input, given that studies in sensory cortices have largely found relatively stable tuning properties. In addition, in association areas, the capability of single-neuron representations to drift could play a role in providing flexibility for coding learned associations[19]. However, our data show that this hypothesis may be overly simplistic, as stability appears to be not only a property of the individual neurons or brain region but also a function of encoded information. As we show in V1, the same neuron can simultaneously exhibit stable tuning properties and more malleable responses to naturalistic stimuli. It remains to be seen if this holds true for other sensory, motor, and association regions as well. Although representational drift will reduce the ability of individual neurons to faithfully encode natural stimuli, there may be compensatory mechanisms for maintaining a stable readout of population activity in the face of ongoing drift[20,36,87].

How would stable tuning properties be maintained in the presence of representational drift? One possibility is that since subnetworks of neurons iso-tuned for orientation exhibit high connectivity (likely established early in development)[47,48,78], fundamental tuning properties would be stabilized throughout the lifetime of the neurons. However, subnetworks of neurons co-responsive to particular time points in a natural stimulus would not necessarily belong to highly connected subnetworks. Although two neurons responding to the same natural stimulus feature might be connected at greater levels than chance[78], they are not necessarily reciprocally connected with other co-responsive neurons in local subnetworks (in the manner of neurons co-tuned for orientation), since there are numerous spatial and temporal features present during each time point of a natural stimulus. For example, if two neurons respond during the same frame of a movie, one neuron might be responding to the spatial frequency of the stimulus while another might be responding to a particular temporal pattern, and these neurons would not necessarily belong to a highly connected subnetwork. This stimulus-dependent difference in reciprocal connectivity is illustrated in Fig. 6e (note that responses to MOV are represented as single response peaks for simplicity). Although we cannot dismiss the possible long-term influence of top-down inputs to V1, this lesser constraint imposed by local connectivity in the case of natural stimuli could contribute to the differential shifts in correlation structure we observed between the two stimuli and the resulting differences in representational drift. Taken together, we propose that co-tuned subnetworks of neurons can preserve fundamental tuning properties while allowing for more flexible

responses to complex naturalistic stimuli. Such a governing principle is potentially applicable to other cortical regions, where highly interconnected neuronal subnetworks may preserve stable encoding of fundamental input features while maintaining flexibility in their responses to more complex inputs.

## Methods

**Animals**. For cortex-wide calcium indicator expression, Emx1-IRES-Cre (Jax Stock #005628) × ROSA-LNL-tTA (Jax Stock #011008) × TITL-GCaMP6s (Jax Stock #024104) triple transgenic mice ($n = 10$) or Slc17a7-IRES2-Cre (Jax Stock #023527) × TITL2-GC6s-ICL-TTA2 (Jax Stock #031562) double transgenic mice ($n = 2$) were bred to express GCaMP6s in cortical excitatory neurons. For interneuron activity measurements, we used GAD2-IRES-Cre (Jax Stock #028867) × TITL2-GC6s-ICL-TTA2 (Jax Stock #031562) double transgenic mice ($n = 4$). For imaging experiments, 12–30 week old mice of both sexes (5 males and 7 females for transgenic excitatory populations, 2 males and 2 females for inhibitory populations) were implanted with a head plate and cranial window and imaged starting 2 weeks after recovery from surgical procedures and up to 10 months after window implantation. The animals were housed on a 12 h light/dark cycle in cages of up to 5 animals before the implants, and individually after the implants. All animal procedures were approved by the Institutional Animal Care and Use Committee at the University of California, Santa Barbara.

**Surgical procedures**. All surgeries were conducted under isoflurane anesthesia (3.5% induction, 1.5–2.5% maintenance). Prior to incision, the scalp was infiltrated with lidocaine (5 mg kg$^{-1}$, subcutaneous) for analgesia, and meloxicam (2 mg kg$^{-1}$, subcutaneous) was administered preoperatively to reduce inflammation. Once anesthetized, the scalp overlying the dorsal skull was sanitized and removed. The periosteum was removed with a scalpel and the skull was abraded with a drill burr to improve the adhesion of dental acrylic.

For standard cranial windows, a 4 mm craniotomy was made over the visual cortex (centered at 4.0 mm posterior, 2.5 mm lateral to Bregma), leaving the dura intact. A cranial window was implanted over the craniotomy and sealed first with silicon elastomer (Kwik-Sil, World Precision Instruments) then with dental acrylic (C&B-Metabond, Parkell) mixed with black ink to reduce light transmission. The cranial windows were made of two rounded pieces of coverglass (Warner Instruments) bonded with a UV-cured optical adhesive (Norland, NOA61). The bottom coverglass (4 mm) fit tightly inside the craniotomy while the top coverglass (5 mm) was bonded to the skull using dental acrylic.

For columnar imaging, we used custom-designed microprisms (Tower Optical) that had a 200 μm square base and a 700 μm right-angle prism (design available on our institutional lab website: https://goard.mcdb.ucsb.edu/resources). The hypotenuse of the right-angle prism was coated with aluminum for internal reflectance. The microprism was attached to a 5 mm diameter coverglass (Warner Instruments) with a UV-cured optical adhesive (Norland, NOA61). Prior to implantation, a 3–4 mm craniotomy was made over the primary visual cortex (centered at 4.0 mm posterior, 2.5 mm lateral to Bregma). A 1 mm length medial-to-lateral incision was then made through the dura and cortex to a depth of 1 mm with a sterilized diamond micro knife (Fine Science Tools, #10100-30) mounted on a manipulator, taking care to avoid blood vessels (~4.0 mm posterior, 2.5 mm lateral to Bregma). Gelfoam (VWR) soaked in sterile saline was used to remove any blood from the incision site. Once the incision site had no bleeding, the craniotomy was submerged in cold sterile saline, and the micro prism was lowered into the cortex using a manipulator, with the imaging face of the prism facing anterior. The micro prisms assembly was completely lowered until the coverglass was flush with the skull, then the edges of the window were sealed with silicon elastomer (Kwik-Sil, World Precision Instruments), then with dental acrylic (C&B-Metabond, Parkell) mixed with black ink. The micro prisms implant enabled imaging from 200 to 900 μm below the coverglass surface, corresponding to ~100–800 μm into the cortex due to approximately 100 μm of dimpling near the top corner of the prism. Cortical layers 2–5 were visible in all recordings, with partial visibility of layer 1 and layer 6. The micro prisms implantations were stable for up to 6 months following the surgery, similar to previously published results[69].

For thinned-skull experiments, a 3–4 mm diameter patch of skull over V1 was carefully thinned using a drill burr and then a rubber polishing bit until fully translucent, being careful to keep the thinned skull wet with sterile saline. After the thinning was complete, the saline was wicked away, a drop of cyanoacrylate (Loctite 406) was placed on the thinned skull and a 3 mm coverglass (Warner) was lowered using a manipulator until flush with the thinned skull. The implant was then sealed with dental acrylic (C&B-Metabond, Parkell) mixed with black ink. The thinned-skull window implantations were stable for up to six months following the surgery, similar to previously published results[88].

After cranial window implantation, a custom-designed stainless steel head plate (eMachineShop.com) was affixed using dental acrylic (C&B-Metabond, Parkell) mixed with black ink. After surgery, mice were administered carprofen (5–10 mg kg$^{-1}$, oral) every 24 h for 3 days to reduce inflammation. The full specifications and designs for head plate and head fixation hardware are available on our institutional lab website (https://goard.mcdb.ucsb.edu/resources).

**Visual stimuli**. All visual stimuli were generated with a Windows PC using MATLAB and the Psychophysics toolbox[89]. Stimuli used for widefield visual stimulation during retinotopic mapping were presented on an LCD monitor (43 × 24 cm, 1600 × 900 pixels, 60 Hz refresh rate) positioned 10 cm from the eye at a 30° angle to the right of the midline, spanning 130° (azimuth) by 100° (elevation) of visual space. For chronic two-photon imaging experiments, visual stimuli were presented on an LCD monitor (17.5 × 13 cm, 800 × 600 pixels, 60 Hz refresh rate) positioned 6 cm from the eye at a 30° angle right of the midline, spanning 120° (azimuth) by 100° (elevation) of visual space. Physical bars affixed to the table and reference point distance measurements were used to ensure that the stimulus monitor was fixed in the exact same location for each experiment.

Retinotopic mapping stimuli consisted of a drifting bar with a contrast reversing checkboard (0.05 cycles degree$^{-1}$ spatial frequency; 2 Hz temporal frequency) that was spherically corrected to account for visual field distortions due to the proximity of the monitor to the mouse's eye. The stimulus was swept in the four cardinal directions, repeated 20–60 times.

For PDG stimulation, 12 full-contrast sine-wave gratings (spatial frequency: 0.05 cycles/deg; temporal frequency: 2 Hz) were presented full-field, ranging from 0° to 330° from vertical in 30° increments. We presented 8 repeats of the drifting grating stimulus; a single repeat of stimulus consisted of all 12 grating directions presented in order for 2 s with a 4 s inter-stimulus interval (gray screen) to allow calcium responses to return to baseline between presentations.

For natural movie visual stimulation, we displayed a grayscale 30 s clip from *Touch of Evil* (Orson Wells, Universal Pictures, 1958) containing a continuous visual scene with no cuts (https://observatory.brain-map.org/visualcoding/stimulus/natural_movies). The clip was contrast-normalized and presented at 30 frames per second. We presented 30 repeats of the natural movie stimulus; each repeat started with 5 s of the gray screen, followed by the 30 s of the movie. The average illuminance of the movie clip is 4.2 lux, with a range of 3–9 lux.

For phase scrambled natural movie visual stimulation, the original 30 s natural movie clip from *Touch of Evil* (Orson Wells, Universal Pictures, 1958) was separated into phase and amplitude spectra using 2-D fast Fourier transform, and its phase structure was randomly scrambled before being reconstructed with the original amplitude data to form the new movie. For the 50% phase scrambled movie, only a random half of the image's phase elements were scrambled. We presented 20 repeats for each of the original natural movie stimulus, the 50% phase scrambled movie, and the 100% phase scrambled movie in random order on every session; each repeat started with 5 s of the gray screen, followed by 30 s of one of the movies. Each version was contrast-normalized and presented at 30 frames per second.

The receptive field mapping stimuli consisted of full-screen length vertical (azimuth) or horizontal (altitude) bars (20° width) with contrast reversing checkerboards (0.04 cycles degree$^{-1}$ spatial frequency; 5 Hz temporal frequency). These bars were presented for 1 s at a number of locations spanning the height (altitude) and width (azimuth) of the screen in random order on every repeat (30 locations for altitude, 40 locations for azimuth) for a total of 10 repeats for each direction, with a 2 s gray screen between repeats.

**Eye tracking**. To confirm that representational drift was not due to eye movements, we performed eye tracking experiments on 4 mice. These mice were head-fixed identically to imaging experiments, but an IR camera (Thorlabs DCC1645C with IR filter removed; Computar T10Z0513CS 5–50 mm f/1.3 lens) was placed such that the image sensor was located immediately lateral to the stimulus monitor. The eye was illuminated with an 850 nm infrared light source to visualize the pupil. Video was acquired at 15 fps and images were analyzed offline in MATLAB (Mathworks).

The pupil was identified for each frame using an automated procedure. In brief, raw images were binarized based on pixel brightness, and the resulting images were morphologically cleaned by removing isolated pixels. For the initial frame, the pupil was manually chosen. For subsequent frames, the pupil was chosen from potential low intensity regions based on a linear combination of size, location, and eccentricity of the pupil in the previous frame.

**Widefield imaging**. After >2 weeks of recovery from surgery, GCaMP6s fluorescence was imaged using a custom widefield epifluorescence microscope[53]. In brief, broad spectrum (400–700 nm) LED illumination (Thorlabs, MNWHL4) was band-passed at 469 nm (Thorlabs, MF469-35), and reflected through a dichroic (Thorlabs, MD498) to the microscope objective (Olympus, MVPLAPO 2XC). Green fluorescence from the imaging window passed through the dichroic and a bandpass filter (Thorlabs, MF525-39) to a scientific CMOS (PCO-Tech, pco.edge 4.2). Images were acquired at 400 × 400 pixels with a field of view of 4.0 × 4.0 mm, leading to a pixel size of 0.01 mm pixel$^{-1}$. A custom light blocker affixed to the head plate was used to prevent light from the visual stimulus monitor from entering the imaging path.

**Widefield post-processing**. Images were acquired with pco.edge camera control software and saved into multi-page TIF files. All subsequent image processing was performed in MATLAB (Mathworks). Visual field sign maps were derived and segmented using established methods[53]. After processing, borders were drawn around each patch, and resulting patches were compared against published sign maps for both size and sign to label each patch as a visual area. Visual areas V1,

LM, AL, PM, LI, RL, and AM were present in all mice. Area V1 was targeted for all further recordings.

**2-Photon imaging.** After >2 weeks' recovery from surgery, GCaMP6s fluorescence was imaged using a Prairie Investigator 2-photon microscopy system with a resonant galvo scanning module (Bruker). For fluorescence excitation, we used a Ti:Sapphire laser (Mai-Tai eHP, Newport) with dispersion compensation (Deep See, Newport) tuned to $\lambda = 920$ nm. Laser power ranged from 40 to 75 mW at the sample depending on GCaMP6s expression levels. Photobleaching was minimal (<1% min$^{-1}$) for all laser powers used. For collection, we used GaAsP photomultiplier tubes (Hamamatsu). A custom stainless-steel light blocker (eMachineShop.com) was mounted to the head plate and interlocked with a tube around the objective to prevent light from the visual stimulus monitor from reaching the photomultiplier tubes. For imaging, we acquired $760 \times 760$ pixel images at 10 Hz using a $16 \times /0.8$ NA microscope objective (Nikon) at fields of view ranging from $690 \times 690$ µm (0.907 µm/pixel) to $414 \times 414$ µm (0.544 µm/pixel). During imaging experiments, the polypropylene tube supporting the mouse was suspended from the behavior platform with high tension springs (Small Parts) to reduce movement artifacts.

For imaging across multiple weeks, imaging fields on a given recording session were manually aligned based on visual inspection of the average map from the reference session recording, guided by stable structural landmarks such as blood vessels and neurons with high baseline fluorescence. Physical controls were used to ensure precise placement of the head plate and the visual stimulus screen relative to the animal, and data acquisition settings were kept consistent across sessions. Recordings were taken once every $7 \pm 1$ days for 5–7 weeks. To acclimate to head fixation and visual stimulus presentation, mice were head-fixed and presented the full series of visual stimuli for 1–2 full sessions prior to the start of their experimental run.

**2-Photon post-processing.** Images were acquired using PrairieView acquisition software and converted into TIF files. All subsequent analyses were performed in MATLAB (Mathworks) using custom code (https://goard.mcdb.ucsb.edu/resources). First, images were corrected for X-Y movement within each session by registration to a reference image (the pixel-wise mean of all frames) using 2-dimensional cross-correlation. Next, to align recordings to the reference session, we used semi-automated rigid registration, similar to prior work[9,90]. First, anchor points were automatically generated from matching image features between average projections detected by the "Speeded-Up Robust Features" (SURF) algorithm (Computer Vision Toolbox, Mathworks). The anchor points were manually corrected through visual inspection, and additional anchor points were added when necessary. These anchor points defined a predicted displacement vector field that would be used to map coordinates from one session to another. For each coordinate, the predicted vector was defined by the average (weighted inversely by distance) of the vectors for all defined anchor points. This vector field was then applied to every frame of the recording to warp the coordinates of each image to the reference coordinate plane.

To identify responsive neural somata, a pixel-wise activity map was calculated using a modified kurtosis measure. Neuron cell bodies were identified using local adaptive threshold and iterative segmentation, using average activity maps across sessions. Individual pixels were filtered with a $3 \times 3$ pixel window before calculating kurtosis to reduce outlier values. Automatically defined ROIs were then manually checked for proper segmentation in a graphical user interface.

To ensure that the response of individual neurons was not due to local neuropil contamination of somatic signals, a corrected fluorescence measure was estimated according to:

$$F_{corrected}(n) = F_{soma}(n) - \alpha(F_{neuropil}(n) - \bar{F}_{neuropil}) \tag{1}$$

where $F_{neuropil}$ was defined as the fluorescence in the region <30 µm from the ROI border (excluding other ROIs) for frame $n$. $\bar{F}_{neuropil}$ is $F_{neuropil}$ averaged over all frames. $\alpha$ was chosen from [0 1] to minimize the Pearson's correlation coefficient between $F_{corrected}$ and $F_{neuropil}$. The $\Delta F/F$ for each neuron was then calculated as:

$$\frac{\triangle F}{F} = \frac{F_n - F_0}{F_0} \tag{2}$$

Where $F_n$ is the corrected fluorescence ($F_{corrected}$) for frame $n$ and $F_0$ is defined as the first mode of the corrected fluorescence density distribution across the entire time series.

**Analysis of 2-photon imaging data.** To minimize potential artifacts introduced by misalignments of the imaging fields across sessions, we manually inspected the average projection and pixel-wise activity maps underlying every defined ROI across all sessions. ROI quality scoring was based on the clarity of the cellular structure in the average fluorescence and activity map (e.g., is the cell structure visible in all weeks and clearly separated from nearby cells) and we included only ROIs of sufficient quality in our analyses (threshold quality of 3 unless indicated otherwise). Briefly, we defined ROI quality as follows: ROIs rated the quality of 4 or 5 were cells that were clearly present across sessions, and the cell structure could be clearly resolved in both the average projection and activity map. ROIs rated the quality of 3 were also cells unambiguously tracked across sessions but had average maps that were often noisier than cells rated 4 or 5 (for example, they may be

identifiable solely by their appearance on the activity map). ROIs rated a quality of 2 were either cells that were not well-tracked or were not unequivocally neuronal somata. ROIs rated the quality of 1 were cells that were not present on the reference session. Each ROI was also marked as either present or not present on each session. Although the activity map was not required to be constant across weeks since cells can change their activity, we were particularly wary of correlated changes between the average fluorescence and activity map that would suggest the movement of the $z$-plane. This process was difficult to accomplish algorithmically, so we used subjective assessments. However, in all cases, we defined ROIs using data spanning both stimulus conditions to eliminate systematic bias.

Reliability on a given session was calculated as the Pearson correlation coefficient (CC) between trial-averaged activity taken from two random halves of trials. For MOV, the 30 trials were randomly subsampled to 8 to match PDG, a CC value was found using this subsampled data, and then this was repeated 10 times and averaged for a final reliability value.

A neuron's responsiveness to a stimulus was determined based on a collective measure of the reliability of the neurons in a given field using time-shuffled data. First, a neuron's activity on each trial was circularly shuffled by a random amount. Next, a reliability value was calculated using this shuffled data. This was repeated 1000 times to yield a distribution of reliability values, and the 99th percentile of this distribution was stored. This 99th percentile threshold was found for every neuron. If a neuron's actual average reliability across sessions was statistically greater than the average of these 99th percentile values (two-tailed one-sample $t$-test), it was classified as responsive to the stimulus. For the experiments in which mice were imaged on only 2 recording sessions, if a neuron's actual average reliability across sessions was at least 3 standard deviations greater than the average of these 99th percentile values, it was classified as responsive to the stimulus. For PDG, a neuron's OSI on a given session was calculated as:

$$OSI = \frac{R_{pref} - R_{orth}}{R_{pref} + R_{orth}} \tag{3}$$

Where $R_{pref}$ is the neuron's average response to its preferred orientation, and $R_{orth}$ is the average response to the orthogonal orientations. To map the preferred orientation for Fig. S2d, a neuron's average orientation response vector was wrapped (averaged between opposite orientations), linearly interpolated onto a 180° scale, and then fit with a gaussian curve to determine peak response.

The RDI with respect to a given session and the reference session was calculated as:

$$RDI = \frac{CC_{ws} - CC_{bs}}{CC_{ws} + CC_{bs}} \tag{4}$$

Where $CC_{ws}$ is the Pearson correlation of the trial-averaged activity of two random halves of trials within a session, and $CC_{bs}$ is the Pearson correlation of the trial-averaged activity of two random halves of trials across the compared sessions. For these calculations, negative CC values were rectified to zero. Control RDI values were calculated by treating half of the trials on the first session as the "test session" and the other half as the 'reference session'.

Visual response events were identified using $\Delta F/F$ data and refined using inferred spike rate data. For each neuron, each frame spanning the length of a stimulus was evaluated for visual response significance by comparing the distribution of its activity values across trials with the distribution of frame-averaged gray-screen period baseline fluorescence values across trials (Wilcoxon sign rank test, right-tailed). Frames were evaluated on a session-by-session basis: if a frame was determined to be significantly responsive in this way on at least two sessions, it was considered to be an event period frame. After each frame was evaluated, the resulting events (periods of contiguous significant frames) were cleaned up by discarding any events consisting of fewer than 5 frames (500 ms) and combining any events 2 or fewer frames (200 ms) apart. Next, the second set of event periods were defined independently using deconvolved spike rate data. A peristimulus time histogram was calculated using spike rate data from all trials across sessions, and frames above a threshold of 2 spikes/s were treated as event periods. To avoid merging distinct response events due to fluorescence tails, the original event periods determined using $\Delta F/F$ data were further refined using these spike rate event periods such that detected $\Delta F/F$ events lasted no longer than 10 frames (1 s) past the end of the leading spike rate event period.

Event types were determined by z-scoring an event's responses on every trial against baseline activity and comparing the distributions of these z-score values between the first two sessions (60 trials) and the last two sessions (60 trials) (two-tailed Wilcoxon rank-sum test). Event instability, or normalized delta z-score, was calculated as:

$$Norm.|\triangle zScore| = \frac{|Z_{late} - Z_{early}|}{Z_{late} + Z_{early}} \tag{5}$$

Where $Z_{early}$ represents the average z-score of an event's trial-averaged activity on the first two sessions and $Z_{late}$ represents the average z-score of an event's trial-averaged activity on the last two sessions. For boxplot comparisons in event analyses, the first quartile of data was compared against the last quartile (Wilcoxon rank-sum test).

To draw boundaries between cortical layers, layer 4 was identified by finding peak ROI density along a user-defined translaminar axis of the window. A 140 µm

window centered at this point was defined as the L2/3/L4 boundary and the L4/L5 boundary, and 150 μm deeper than the L4/L5 boundary was defined as the L5/L6 boundary[91].

For comparing pupil size to visual responsiveness, a $\Delta F/F$ gain factor was calculated for every trial. For each trial, a linear least-squares fit was performed between each neuron's activity for the trial and its trial-averaged activity for the given session, and the alpha values produced by these fits were averaged across neurons to yield the trial's gain factor.

Pairwise signal correlations were calculated as the Pearson correlation coefficient between two neurons' trial-averaged activity on a given session. For pairwise noise correlations, a neuron's trial-average activity was first subtracted from its activity on every trial. All frames were then concatenated into a continuous signal, and the Pearson correlation coefficient was found between these full traces (for MOV, only the first 5760 frames were used to match the length of the PDG recording). Calculation of noise correlations using the average correlation of response vectors across stimulus presentations produced similar results[36].

**Statistical information.** To test the statistical significance of single groups compared to a group of zero mean, one-sample *t*-tests (normally distributed data) or Wilcoxon signed-rank (non-parametric) tests were performed. For comparing experimental groups, two-sample paired *t*-tests were performed for paired groups, and either unpaired *t*-tests (normally distributed data) or Wilcoxon rank-sum (non-parametric) tests were performed for unpaired groups. Nested data were compared using a linear mixed-effects model (fixed effect for stimulus, random effect for mouse). Two-way ANOVA was performed for testing the effects of multiple factors. All tests were performed two-tailed unless indicated otherwise.

**Reporting summary.** Further information on research design is available in the Nature Research Reporting Summary linked to this article.

## Data availability

The processed neuronal response data (Figs. 1–6) generated in this study have been deposited in the Dryad database (https://doi.org/10.25349/D9M606). Source data are provided with this paper.

## Code availability

All of the code used for analysis are available on GitHub (https://github.com/ucsb-goard-lab/Representational-Drift-Project).

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

## Acknowledgements

We would like to thank Ji Xia and Ralf Wessel for their comments on the manuscript. This work was supported by grants to M.J.G. from the Whitehall Foundation, Larry L. Hillblom Foundation, NSF (1707287), and NIH (R00 MH104259).

## Author contributions

T.D.M. and M.J.G. designed the experiments; T.D.M. conducted the experiments and analyzed the data, with guidance from M.J.G.; T.D.M. and M.J.G. wrote the manuscript.

## Competing interests

The authors declare no competing interests.
