## [Peer Review File · Nature Communications]

Stimulus-dependent representational drift in primary visual cortexREVIEWER COMMENTS

Reviewer #1 (Remarks to the Author):

The study by Marks and Goard addresses the potential stability of visually-evoked responses in the mouse visual cortex, comparing activity evoked by either traditional drifting gratings or natural scene movies. They find that, especially for movies, neural responses exhibit "representational drift" across sessions. They also find changes in the correlational structure of local networks, although they do not see differences by cell type or layer. However, the overall conclusions are quite limited - there is little exploration of what aspects of the visual stimulus or behavioral state drive the changes, what the cellular or circuit mechanism might be, nor what consequences for perception might follow. These gaps are particularly evident given the significant body of work that already exists on this topic. In addition, there are some analytical issues that make it difficult to fully interpret the results.

1. The question - the stability of sensory representations - is a well examined one. Moreover, despite the lengthy introduction, many earlier studies are not mentioned by the authors (Gavornik and Bear 2014, Makino and Komiyama, Puscian...Higley 2020) that explicitly address the relationship between repeated stimulus presentation and response properties. In particular, Puscian et al show that repeated presentation of drifting gratings drives changes in response amplitude without alteration in sensory representation.

2. The authors use correlations in activity across entire trials as a metric for both reliability and responsiveness, as well as the readout of "representational drift". However, it is unclear how to interpret this signal, as it will be a mixture of both visually-evoked signals and spontaneous activity from inter-stimulus intervals. For example, relative increases in the amount of spontaneous fluctuations will necessarily reduce the correlation between trials in the absence of any change in the "visually evoked response". While this type of overall correlation is potentially helpful for calcium imaging data that is already a filtered version of the underlying spike train, the lack of similarity to other studies and the potential confound from inter-stimulus periods limits interpretation.

3. A related issue is the direct comparison of responses evoked by brief gratings versus natural scenes. The gratings have clear onset times, and therefore robust, discrete onset transients of large amplitude that are clearly seen in the data (Figure 1F). In contrast, natural scenes may or may not have such robust and repeatable onset transients, leading to "muddier" responses whose signal-to-noise may be more sensitive to spontaneous fluctuations (also apparent in Figure 1F). Thus, the selective "representational drift" may simply reflect the apples-to-oranges comparison of the two stimulus sets and an unconventional response metric rather than a fundamental difference in plasticity.

4. The authors provide some analysis of changes in arousal across training by measuring pupil diameter. However, the demonstration that average pupil diameter does not correlate over sessions with RDI seems cursory at best. It would be helpful to see that the pupil changes observed do provide within-session information that tracks visual responsiveness, as published previously by many groups. In addition, other behavioral state variables (e.g., whisking) might provide an alternative metric. This seems particularly critical, as the authors in the end find no convincing explanation for the RDI.

5. Finally, in the absence of a mechanism or function, it would be interesting to at least see greater exploration of the relationship between visual stimulation and RDI. For example, does RDI occur for movies if only gratings are presented from weeks 2-6? Does more frequent stimulus presentation drive faster or more robust RDI? Does the RDI persist if visual stimulation is stopped for a period of time?

Reviewer #2 (Remarks to the Author):

The manuscript examines stability of visual cortex responses to drifting gratings and natural movies over the course of several weeks of recording. The stability of neuronal representations and how they change over time is an important question as it constrains for how these representations can influence downstream structures and guide behavior.

The primary claims of the paper are as follows:

- (i) responses to natural movies are less stable than responses to drifting gratings;
- (ii) this difference cannot be trivially explained by differences in the magnitude of response events;
- (iii) changes in responses cannot be explained by behavioral variables;
- (iv) representational drift appears to affect different layers as well as both excitatory and inhibitory neurons to a similar degree.

Overall, this is an interesting and well-executed study. The fact that natural movie responses exhibit a greater degree of plasticity is clear. What is less clear is what may be driving this difference and whether behavioral changes might contribute to apparent representational drift, contrary to the claims of the manuscript.

1. It is clear that natural movie responses are less stable than that to drifting gratings but it is not clear what is it about natural movies that underlies this difference. As the result, while this is an interesting observation, the reader is left wondering what it might imply about neural circuit function. Are MOV stimuli simply richer and more diverse, and therefore more likely to reveal representational drift over sessions? If that were the case, it would not decrease my enthusiasm for the paper, as it would suggest that simple parametric stimuli that are typically used overestimate stability of responses.

The authors speculate that differences in connectivity of ensembles recruited by gratings vs movies may be responsible. However, the speculation that neurons that happen to respond at the same moment during presentation of the MOV stimulus are less likely to be integrated into the same synaptic subnetwork than those that respond to the same grating orientation is at odds with prior work. Cossell et al. (manuscript ref. 72) showed that neurons responses to natural images were highly predictive of their synaptic connectivity. While that study did not use natural movies, it would be surprising if the results did not also extend to movie stimuli. Therefore, it seems likely that neuronal ensembles that are co-active during natural movie presentation are also recurrently connected.

2. The data supporting the claim that behavioral variables cannot explain differential stability of PDG and MOV responses are weak. Firstly, the authors do show that pupil diameter decreases over the time. The corresponding decrease in arousal may cause the overall reduction in the strength response events for both PDG and MOV stimuli (Figure 2D). Could MOV responses be more sensitive to animal's level of arousal?

The authors state that "pupil size stabilized after the third session, and is unlikely to contribute to progressive drift observed in later sessions". The only corresponding data figure is Supplemental Figure 8C. It would be much easier to relate this to RDI measurements if pupil size was plotted for each recording week (as RDI values in the main figures). Since there is a lot of mouse-to-mouse variability in RDI curves and pupil size was only measured in 4 of the mice, it would also be helpful to compare pupil measurements and response stability for those specific animals.

Secondly, pupil diameter and position are the only aspects of behavior quantified. No videography or other data is presented. Without additional behavioral measurements, I would encourage the authors to carefully consider the claims they make about potential behavior-related confounds and perhaps focus on arousal (based on pupillometry) rather than behavior in general.

3. I am not sure what additional insights are gained from the signal correlation analysis in Figure 6. Surely, if single cell trial-average responses change, signal correlations are likely to change too. It would have been quite surprising if single cell responses changed in such a way as to preserve signal

correlations. Am I missing something here?

4. An important question that is not discussed in the manuscript is whether repeated presentation of the same stimulus may be contributing to representational drift. Would the same amount of drift be observed between week 1 and week 7 if the stimulus was not presented on weeks 2-6, or if it was presented with fewer repetitions? If the authors' data can shed light on this question, it would strengthen the manuscript. Otherwise, the authors ought to consider the potential role of repeated exposure to the stimulus in the discussion.

5. Animals of a wide range of ages (12 – 30 weeks) are used in the study. Does animal age explain any of the variability in the extent of representational drift?

Reviewer #3 (Remarks to the Author):

The manuscript by Mark and Goard titled 'Stimulus-dependent representational drift in primary visual cortex' examines the stability of natural scene representation in primary visual cortex (V1) relative to the stability of simpler grating stimuli. This is an important issue to address because although it is well recognized that orientation tuning to classic Cartesian stimuli in the upper layers of primary cortex is stable across days, it is completely unknown whether responses to complex, naturalistic stimuli are stable. The study is comprehensive and uses the micro-prism technique to capture neural responses from layers 2/3, 4, and 5. The scholarship of the manuscript is excellent; the manuscript is well-written and was a pleasure to read. The quality of the data set is superb and will likely serve as reference material for years to come. Associating a metric of quality of tracking single neurons across sessions is extremely useful. The discovery that the representation of naturalistic movies in V1 is less stable than classic oriented stimuli is of interest to the fields of mammalian vision and sensory systems in general.

Major

1. The authors include a results section on the topic of whether the greater instability observed for MOV can be accounted for by behavioral state. This is important. It would be useful to examine this issue in further detail. Eye movements: do the eyes move more during MOV than PDG for the four mice examined? Does this change with sessions number? Cumulative distribution of pupil position for each individual mouse for all sessions could address this issue (ideally degrees, but relative to median with-session ok)

2. If possible, it would be valuable perform additional analysis on the inhibitory neuron population, specifically, to sub-divide the inhibitory neurons into functional classes. For example, are the inhibitory neurons broadly tuned for orientation more or less stable than inhibitory neurons sharply tuned for orientation?

3. The authors' justification for concluding that their data suggest local network connectivity may influence stability is tenuous. Do the authors have any data demonstrating that instability of naturalistic images is not the result influence arising from top-down input? ...if not, fine, then remove this claim. Neither the main conclusion, nor the significance of the manuscript is altered if this statement is removed or moved to the Discussion.

4. It would be useful if the description of the segmentation included more details. (I) What parameters and range are used to modify the kurtosis measure? What is the median adaptive threshold, is the spatial extent of the localization fixed, and what range of threshold are typically used? (II) During the manual check (which is a good idea), what are the criteria for rejection of a segment? Inclusion criteria? (the graphical interface is a nice touch, that should help ensure consistency across users)

Minor

Please report the luminance of each frame in the movie sequence.

Line 75. The work cited is not peer reviewed, so using this work as a rationale for further study is problematic. Furthermore, the rationale is sufficiently strong without including a reference to results uploaded to BioArchive.

Line 239. The term 'highly salient' features should be avoided when interpreting the data generated here. There is no salience to the stimuli- if anything, the animal may be learning to ignore these stimuli, given the presented stimuli do not have behavioral significance. The authors' point being made here is unclear.

Figure1 Legend. Please do not use term the term 'watching'.

Line 629. How many pixels typically went into calculating the neuropil per neuron? What are the spatial dimensions of a pixel?

Response to Reviewers

We would like to thank the reviewers for their insightful comments on the manuscript. We have added several additional experiments and revised the manuscript to address the concerns raised in the reviews. Indicated changes to the manuscript are marked in red and line numbers are specified in the response.

Reviewer #1 (Remarks to the Author):

The study by Marks and Goard addresses the potential stability of visually-evoked responses in the mouse visual cortex, comparing activity evoked by either traditional drifting gratings or natural scene movies. They find that, especially for movies, neural responses exhibit "representational drift" across sessions. They also find changes in the correlational structure of local networks, although they do not see differences by cell type or layer. However, the overall conclusions are quite limited - there is little exploration of what aspects of the visual stimulus or behavioral state drive the changes, what the cellular or circuit mechanism might be, nor what consequences for perception might follow. These gaps are particularly evident given the significant body of work that already exists on this topic. In addition, there are some analytical issues that make it difficult to fully interpret the results.

We thank the reviewer for their comments, though we are not sure if we understand the primary criticism. A significant portion of our manuscript concentrated on aspects of the stimulus (Figures 2, 5, S9), behavioral state (Figures S8-9), and cell types/layers (Figures 3, 4) that might be responsible for the differential representational drift. Admittedly, a number of the findings were negative results (e.g., the differential drift could not be explained by differences across layers or higher order spatial correlations). However, checking these possibilities is an important part of investigating the mechanism. We also provide a hypothesis for explaining the differential drift - namely, that subnetworks of highly connected neurons with similar stimulus tuning stabilize responses (Figures 6, S10). This allows the same neurons to reliably encode particular visual features such as orientation, while responses to other visual features are more malleable.

We agree that there is already a significant body of work on representational drift, but this is the first manuscript that describes differential representational drift within the same neurons for different stimuli. Although there remain questions on the underlying mechanism, this finding alone changes our understanding of representational drift.

1. The question - the stability of sensory representations - is a well examined one. Moreover, despite the lengthy introduction, many earlier studies are not mentioned by the authors (Gavornik and Bear 2014, Makino and Komiyama, Puscian...Higley 2020) that explicitly address the relationship between repeated stimulus presentation and response properties. In particular, Puscian et al show that repeated presentation of drifting gratings drives changes in response amplitude without alteration in sensory representation.

This is an important distinction - these studies (and a number of others) are focused on perceptual learning rather than on representational drift. We know that perceptual learning can change cortical representations (even for orientation), but this is very different from spontaneous changes due to representational drift. We have clarified this point in the manuscript (lines 208-210).

That said, it is possible that the repeated stimulus alone induces some perceptual learning, despite the lack of task or reward (this point was also raised by Reviewer #2). Indeed, this was found in the Gavornik & Bear 2014 paper upon repeated passive presentation, albeit with a simpler stimulus sequence and more repetitions (800 repeats) presented on a daily basis. To address this possibility, we have run experiments to test representational drift across the same time span, but without the intervening stimulus presentations (Figure S8; lines 210-218; specific experiments explained in point 5 below).

2. The authors use correlations in activity across entire trials as a metric for both reliability and responsiveness, as well as the readout of "representational drift". However, it is unclear how to interpret this signal, as it will be a mixture of both visually-evoked signals and spontaneous activity from inter-stimulus intervals. For example, relative increases in the amount of spontaneous fluctuations will necessarily reduce the correlation between trials in the absence of any change in the "visually evoked response". While this type of overall correlation is potentially helpful for calcium imaging data that is already a filtered version of the underlying spike train, the lack of similarity to other studies and the potential confound from inter-stimulus periods limits interpretation.

The reason our metric uses a between-trial correlation based metric instead of a tuning curve similarity based metric is that we do not know exactly which features are inducing reliable responses in the natural movies. With the PDG stimulus, we do show equivalence between our between-trial metric and orientation tuning metrics (Figure S2). However, with the MOV

stimulus, we believe our representational drift metric is the best method for determining how responses change over time. Note that our event-based metric (Figure 2) also revealed similar changes despite analyzing changes in single response events.

However, we agree with the reviewer that inter-stimulus intervals may influence our metric. To address the concern about inter-stimulus intervals, we carried out additional experiments and analyses. Specifically, we checked the stability of responses to PDG and MOV stimuli using matched temporal structures, and found that eliminating the interstimulus intervals from the PDG stimulus (or adding interstimulus intervals to the MOV stimulus) did not affect the differential representational drift (Figure S7; lines 187-208; specific experiments described in point 3 below).

3. A related issue is the direct comparison of responses evoked by brief gratings versus natural scenes. The gratings have clear onset times, and therefore robust, discrete onset transients of large amplitude that are clearly seen in the data (Figure 1F). In contrast, natural scenes may or may not have such robust and repeatable onset transients, leading to "muddier" responses whose signal-to-noise may be more sensitive to spontaneous fluctuations (also apparent in Figure 1F). Thus, the selective "representational drift" may simply reflect the apples-to-oranges comparison of the two stimulus sets and an unconventional response metric rather than a fundamental difference in plasticity.

We agree with this concern and have imaged several additional mice to address this issue. In these new recordings, we matched the temporal structure of the stimulus presentations between PDG and MOV stimuli. We found that PDG and MOV stimuli with matched temporal structure exhibit similar differential representational drift as the original stimuli (lines 187-208; Figure S7, reprinted below).

Supplementary Figure 7: Matching temporal structure between stimuli does not eliminate RDI differences.

(a) Schematic of visual stimuli. The PDG continuous stimulus was designed to have similar temporal structure to the MOV stimulus; the MOV discrete stimulus was designed to have similar temporal structure to the PDG stimulus.

(b) Example neuron responses to each of the stimuli. Recording sessions are separated by white lines.

(c) Field-average RDI curves for the original PDG and MOV stimuli ($n = 449, 447, 441, 402, 382, 329$ neurons in 5, 5, 5, 4, 4, 3 imaging fields for sessions 2-7 respectively). Error bars are \pm s.e.m. MOV is significantly different from PDG ($F_{1,898} = 8.7, F_{1,894} = 27.0, F_{1,882} = 11.8, F_{1,804} = 13.2, F_{1,764} = 26.8, F_{1,658} = 34.6$ for sessions 2-7 respectively; $**p < 0.01, ***p < 0.001$; linear mixed-effects model, fixed effect for stimulus, random effect for mouse).

(d) Field-average RDI curves for the temporal-structure-matched stimuli ($n = 449, 447, 441, 402, 382, 329$ neurons in 5, 5, 5, 4, 4, 3 imaging fields for sessions 2-7 respectively). Error bars are \pm s.e.m. MOV discrete is significantly different from PDG continuous ($F_{1,898} = 4.8, F_{1,894} = 24.9, F_{1,882} = 24.0, F_{1,804} = 18.8, F_{1,764} = 16.2, F_{1,658} = 20.0$ for sessions 2-7 respectively; $*p < 0.05, ***p < 0.001$; linear mixed-effects model, fixed effect for stimulus, random effect for mouse).

4. The authors provide some analysis of changes in arousal across training by measuring pupil diameter. However, the demonstration that average pupil diameter does not correlate over sessions with RDI seems cursory at best. It would be helpful to see that the pupil changes observed do provide within-session information that tracks visual responsiveness, as published previously by many groups. In addition, other behavioral state variables (e.g., whisking) might provide an alternative metric. This seems particularly critical, as the authors in the end find no convincing explanation for the RDI.

We have divided our previous supplementary figure on eye movements and pupil dilation (formerly Figure S8) into two supplementary figures to address this and other concerns raised by the reviewers (Figures S10 and S11). We have added analysis showing that pupil dilation is moderately correlated with response magnitude, as shown by previous papers, and that neither eye movements or pupil size appreciably differ across stimulus conditions (lines 290-292; 767-770). We also analyzed eye movements and pupil dilation as a function of session number, and added all results to the paper (lines 284-302, Figures S10 and S11).

However, our principal finding is just what the reviewer described - while we observed some eye movements and changes in pupil dilation, since the changes are similar across stimulus sets, the behavioral variables are not sufficient to explain the differential drift between PDG and MOV stimuli.

5. Finally, in the absence of a mechanism or function, it would be interesting to at least see greater exploration of the relationship between visual stimulation and RDI. For example, does RDI occur for movies if only gratings are presented from weeks 2-6? Does more frequent stimulus presentation drive faster or more robust RDI? Does the RDI persist if visual stimulation is stopped for a period of time?

To address this issue, we imaged at Day 0 to record baseline responses to PDG and MOV stimulus, then imaged during a single session on Day 42 without any visual stimulus presentations in the interim. We chose not to present either stimulus in the interim in order to be able to directly compare the representational drift across stimuli. We found that even in the absence of extensive repeated stimulus presentation a period of 42 days produces differential representational drift between the stimuli. We have added these results to the manuscript (lines 210-218; Figure S8, reprinted below). The difference in RDI between stimuli qualitatively appeared smaller for the spaced imaging sessions, though it is difficult to make any firm conclusions given the smaller sample size of mice. Regardless, the repeated presentation cannot explain the difference in representational drift we see across visual stimuli.

Supplementary Figure 8: Representational drift persists in the absence of repeated stimulus presentation.

(a) Responses to PDG and MOV stimuli on D0 and D42 for two example neurons (top and bottom). White lines separate recording sessions.

(b) Average D42 RDI values ($n = 115$ neurons from 3 imaging fields) compared to the original average RDI curves from Fig. 1i (desaturated curves). Error bars are \pm s.e.m. MOV RDI is significantly different from PDG RDI ($F_{1,228} = 11.6$, $***p < 0.001$; linear mixed-effects model, fixed effect for stimulus, random effect for mouse).

Reviewer #2 (Remarks to the Author):

The manuscript examines stability of visual cortex responses to drifting gratings and natural movies over the course of several weeks of recording. The stability of neuronal representations and how they change over time is an important question as it constrains for how these representations can influence downstream structures and guide behavior.

The primary claims of the paper are as follows:

- (i) responses to natural movies are less stable than responses to drifting gratings;

(ii) this difference cannot be trivially explained by differences in the magnitude of response events;

(iii) changes in responses cannot be explained by behavioral variables;

(iv) representational drift appears to affect different layers as well as both excitatory and inhibitory neurons to a similar degree.

Overall, this is an interesting and well-executed study. The fact that natural movie responses exhibit a greater degree of plasticity is clear. What is less clear is what may be driving this difference and whether behavioral changes might contribute to apparent representational drift, contrary to the claims of the manuscript.

We thank the reviewer for their positive comments and have addressed their concerns about the behavioral changes in the revised manuscript.

1. It is clear that natural movie responses are less stable than that to drifting gratings but it is not clear what is it about natural movies that underlies this difference. As the result, while this is an interesting observation, the reader is left wondering what it might imply about neural circuit function. Are MOV stimuli simply richer and more diverse, and therefore more likely to reveal representational drift over sessions? If that were the case, it would not decrease my enthusiasm for the paper, as it would suggest that simple parametric stimuli that are typically used overestimate stability of responses.

We also suspected that the difference may be due to the “complexity” of the MOV stimulus in comparison to the PDG stimulus. However, we were surprised that phase scrambling of the MOV stimuli had no effect on the progressive representational drift (Figure 5). Even when the stimulus was 100% phase scrambled (preserving only spatial and temporal frequency distribution), the representational drift was as pronounced as for the unscrambled movie (Figure 5b). This indicates that higher order spatial features alone are not responsible for the difference between stimuli.

The authors speculate that differences in connectivity of ensembles recruited by gratings vs movies may be responsible. However, the speculation that neurons that happen to respond at the same moment during presentation of the MOV stimulus are less likely to be integrated into the same synaptic subnetwork than those that respond to the same grating orientation is at odds with prior work. Cossell et al. (manuscript ref. 72) showed that neurons responses to

natural images were highly predictive of their synaptic connectivity. While that study did not use natural movies, it would be surprising if the results did not also extend to movie stimuli. Therefore, it seems likely that neuronal ensembles that are co-active during natural movie presentation are also recurrently connected.

Yes, this is an important point that we should have explained more clearly. We are not arguing that neurons with similar responses to MOV stimuli are never strongly connected. However, with PDG stimuli, neurons responding to a particular grating are all highly connected within a local subnetwork of iso-oriented neurons, which constrains representational drift. On the other hand, two neurons responding to the same time point in the MOV stimuli may be highly connected, but they are not necessarily reciprocally connected with other neurons in a local subnetwork (since all of the responsive neurons may be responding to different spatial and temporal aspects of the stimulus). As a result, we expect the responses of the neurons will be less constrained by local connectivity. We have discussed this point in more detail in the discussion (lines 481-484).

2. The data supporting the claim that behavioral variables cannot explain differential stability of PDG and MOV responses are weak. Firstly, the authors do show that pupil diameter decreases over the time. The corresponding decrease in arousal may cause the overall reduction in the strength response events for both PDG and MOV stimuli (Figure 2D). Could MOV responses be more sensitive to animal's level of arousal?

Good point. One possibility is that the different stimuli actually cause differential changes in arousal. To test this, we analyzed the change in pupil diameter separately for each stimulus, and found that changes in average pupil size were not statistically different across stimulus conditions. The results have been added to the manuscript (lines 290-302, Figure S11). Furthermore, we observe differential representational drift between stimuli even when we only show the mice the stimuli during two sessions spaced by 42 days (Figure S8), when changes in arousal due to repeated stimuli are unlikely to influence responses.

The authors state that “pupil size stabilized after the third session, and is unlikely to contribute to progressive drift observed in later sessions”. The only corresponding data figure is Supplemental Figure 8C. It would be much easier to relate this to RDI measurements if pupil size was plotted for each recording week (as RDI values in the main figures). Since there is a lot of mouse-to-mouse variability in RDI curves and pupil size was only measured in 4 of the mice,

it would also be helpful to compare pupil measurements and response stability for those specific animals.

We have plotted RDI curves and pupil size measurements for both stimuli for each of the four mice in which these were simultaneously recorded (Figures S10, S11). We found no clear relationship between either changes in eye movements or changes in pupil size and the stability of a mouse's responses to either stimulus.

Secondly, pupil diameter and position are the only aspects of behavior quantified. No videography or other data is presented. Without additional behavioral measurements, I would encourage the authors to carefully consider the claims they make about potential behavior-related confounds and perhaps focus on arousal (based on pupillometry) rather than behavior in general.

Agreed, we have corrected the text (lines 276, 299-302).

3. I am not sure what additional insights are gained from the signal correlation analysis in Figure 6. Surely, if single cell trial-average responses change, signal correlations are likely to change too. It would have been quite surprising if single cell responses changed in such a way as to preserve signal correlations. Am I missing something here?

This is an important point, and we realize we were not sufficiently clear in the original manuscript. Single cell response changes do not necessarily imply a change in the signal correlations. For example, if several neurons change their responses at a certain time point in the movie in the same direction, that would result in representational drift without any effect on the signal correlation structure. Indeed, if the representational drift were purely due to gain changes (e.g., from habituation to familiar stimuli or changes in arousal), then we would expect correlated changes in response, resulting in representational drift without changes in the signal correlation structure. Instead, what we find is that neurons exhibit bidirectional changes in individual response events (Figure 2) that are independent of changes in other neurons (Figure 6). We have added additional discussion of these results (lines 385-388).

4. An important question that is not discussed in the manuscript is whether repeated presentation of the same stimulus may be contributing to representational drift. Would the same amount of drift be observed between week 1 and week 7 if the stimulus was not presented on weeks 2-6, or if it was presented with fewer repetitions? If the authors' data can shed light on

this question, it would strengthen the manuscript. Otherwise, the authors ought to consider the potential role of repeated exposure to the stimulus in the discussion.

We agree that this is a potential concern (this was also pointed out by reviewer #1). To address this concern, we carried out a new set of experiments in which mice were only imaged at D0 and D42 without any intervening stimulus presentations. We found that even in the absence of extensive repeated stimulus presentation a period of 42 days produces differential representational drift between the stimuli. We have added these results to the manuscript (lines 208-218, Figure S8).

5. Animals of a wide range of ages (12 - 30 weeks) are used in the study. Does animal age explain any of the variability in the extent of representational drift?

To check whether animal age had any affect on representational drift, we analyzed RDI as a function of animal age at the first imaging session. We found a trend toward lower RDI in older animals, but no significant correlation for either stimulus. We have added this finding to the manuscript (lines 270-271, Figure S5d).

Reviewer #3 (Remarks to the Author):

The manuscript by Mark and Goard titled 'Stimulus-dependent representational drift in primary visual cortex' examines the stability of natural scene representation in primary visual cortex (V1) relative to the stability of simpler grating stimuli. This is an important issue to address because although it is well recognized that orientation tuning to classic Cartesian stimuli in the upper layers of primary cortex is stable across days, it is completely unknown whether responses to complex, naturalistic stimuli are stable. The study is comprehensive and uses the micro-prism technique to capture neural responses from layers 2/3, 4, and 5. The scholarship of the manuscript is excellent; the manuscript is well-written and was a pleasure to read. The quality of the data set is superb and will likely serve as reference material for years to come. Associating a metric of quality of tracking single neurons across sessions is extremely useful. The discovery that the representation of naturalistic movies in V1 is less stable than classic oriented stimuli is of interest to the fields of mammalian vision and sensory systems in general.

We thank the reviewer for the kind comments on the manuscript.

Major

1. The authors include a results section on the topic of whether the greater instability observed for MOV can be accounted for by behavioral state. This is important. It would be useful to examine this issue in further detail. Eye movements: do the eyes move more during MOV than PDG for the four mice examined? Does this change with sessions number? Cumulative distribution of pupil position for each individual mouse for all sessions could address this issue (ideally degrees, but relative to median with-session ok)

This is a good question (related to a question raised by reviewer #2). We analyzed the change in eye movements and pupil diameter separately for each stimulus, and found that changes in these factors were not significantly different between stimuli. We also analyzed changes across session number and found no clear relationship between changes in either eye movements or pupil size and the stability of a mouse's responses to either stimulus. The results have been added to the manuscript (lines 284-302, Figures S10 and S11).

2. If possible, it would be valuable perform additional analysis on the inhibitory neuron population, specifically, to sub-divide the inhibitory neurons into functional classes. For example, are the inhibitory neurons broadly tuned for orientation more or less stable than inhibitory neurons sharply tuned for orientation?

To address this, we have divided neurons into sharply-tuned and broadly-tuned groups (threshold OSI = 0.4 based on distribution, see Reviewer Figure 1 inset). We found that RDI may be slightly higher for sharply-tuned interneurons compared to broadly-tuned interneurons (see Reviewer Figure 1 below). These findings are potentially interesting, as they indicate that subsets of neurons (presumably the more broadly-tuned PV+ interneurons) may have lower levels of representational drift. However, although differences in tuning selectivity have been observed across genetically-defined inhibitory subtypes, we do not feel confident classifying inhibitory neuron subtypes based only on tuning selectivity without confirming using subtype-specific driver lines, so we decided to leave this as a reviewer figure.

Reviewer Figure 1. Representational drift across stimuli for sharply- and broadly- tuned inhibitory interneurons. Plot of RDI over time across stimuli for broadly-tuned neurons (PDG: blue; MOV: magenta) and sharply-tuned neurons (PDG: cyan; MOV: beige). Inset, distribution of OSIs for inhibitory interneurons; grey dashed line in histogram indicates threshold for sharply-tuned (OSI > 0.4) versus broadly-tuned (OSI < 0.4) neurons. Asterisks indicate significant differences for each session for sharp vs. broad within each stimulus (MOV: beige, PDG: cyan, * $p < 0.05$, Wilcoxon rank sum test).

3. The authors' justification for concluding that their data suggest local network connectivity may influence stability is tenuous. Do the authors have any data demonstrating that instability of naturalistic images is not the result influence arising from top-down input? ...if not, fine, then remove this claim. Neither the main conclusion, nor the significance of the manuscript is altered if this statement is removed or moved to the Discussion.

We agree - although we observe local changes in signal correlations, more distal top-down inputs could also be influencing responses. We have removed this statement from the manuscript and discussed it with more care in the Discussion (lines 484-487).

4. It would be useful if the description of the segmentation included more details. (I) What parameters and range are used to modify the kurtosis measure? What is the median adaptive threshold, is the spatial extent of the localization fixed, and what range of threshold are typically used? (II) During the manual check (which is a good idea), what are the criteria for rejection of a segment? Inclusion criteria? (the graphical interface is a nice touch, that should help ensure consistency across users)

We have added more details to the methods to describe the ROI segmentation (lines 672-673; 691-693; 702-707). Briefly: (I) We filtered the individual pixels with a 3 x 3 pixel window before calculating kurtosis to reduce outlier values. (II) The scoring was based on the clarity of the cellular structure in the average fluorescence and activity map (e.g., is the cell structure visible in all weeks and clearly separated from nearby cells). Although the activity map was not required to be constant across weeks since cells can change their activity, we were particularly wary of correlated changes between the average fluorescence and activity map that would suggest movement of the z-plane. This process was difficult to accomplish algorithmically, so we used subjective assessments. However, in all cases we defined ROIs using data spanning both stimulus conditions to eliminate systematic bias. In addition, we checked that our quality score inclusion threshold did not affect the key results (Figure S1f).

Minor

Please report the luminance of each frame in the movie sequence.

We have added information on the luminance to the Methods (line 586).

Line 75. The work cited is not peer reviewed, so using this work as a rationale for further study is problematic. Furthermore, the rationale is sufficiently strong without including a reference to results uploaded to BioArchive.

We have deleted this sentence from the rationale.

Line 239. The term 'highly salient' features should be avoided when interpreting the data generated here. There is no salience to the stimuli- if anything, the animal may be learning to ignore these stimuli, given the presented stimuli do not have behavioral significance. The authors' point being made here is unclear.

Sorry, we should have clarified that we were referring to bottom-up salience (e.g., sudden movements or changes in luminance, agnostic to behavioral significance). We wanted to investigate whether movie frames with high bottom-up salience were driving high magnitude / stable responses in substantial subsets of cells, but our data indicated the opposite. We have clarified this point in the manuscript (lines 272-273).

Figure1 Legend. Please do not use term the term 'watching'.

Thank you, corrected.

Line 629. How many pixels typically went into calculating the neuropil per neuron? What are the spatial dimensions of a pixel?

The neuropil annulus varied depending on the local cell density (we excluded other ROIs from the neuropil mask), but the average size was approximately 30 pixels. The spatial dimension of each pixel is 0.545 x 0.545 um for a typical recording (for the 2x magnification used in most experiments). We have added the details on the pixel size of the images to the methods (lines 641-643).

Note on change to statistical tests: During the revision, we realized that in some cases our data had a nested structure (a large sample of neurons from a smaller number of mice). Since there appears to be between-mouse differences in RDI effects (Figure S5), counting each neuron as an independent sample could inflate Type I errors (Aarts et al., *Nat. Neurosci.*, 2014). To address this, nested data were compared using a linear mixed-effects model (fixed effect for stimulus, random effect for mouse; see lines 156-157; 783-784). This did not affect the statistical significance of any of the key results.

REVIEWER COMMENTS

Reviewer #1 (Remarks to the Author):

1. Prior work on variable sensory responses over time was not limited to perceptual learning. Puscian et al. 2020 showed that passive presentation of visual stimuli in the absence of any task was sufficient to drive significant reduction in visual responses, with ~70 stimuli per day. Recent work from the Axel lab also demonstrates in the olfactory system that passive stimulus presentation is associated with representational drift. These earlier findings should be emphasized in the introduction to not give the sense that the present manuscript is the first to show such longitudinal variation.

2. The authors have done a reasonable job responding to many of the issues I raised (some in common with other reviewers). My major outstanding concern, is what the RDI is really measuring. It is unlike other standard metrics (peak response amplitude or integral, tuning properties, accurate representation of the stimulus, etc). It seems to me that the correlation in activity during stimulus presentation across trials/sessions (i.e., the RDI) has the potential to be confounded by changes in response amplitude, kinetics, and relative weight in comparison to ongoing stimulus-unrelated activity. None of these varying contributors are really fleshed out here, leaving me unsure how to interpret the results. What does a change in RDI "mean" other than a reduction in correlation of signal? Is it more difficult to predict that a stimulus occurred? Is it more difficult to predict features of the stimulus from the recorded activity? I believe the authors need to do a much more thorough job working through these ideas in the discussion at least, in the absence of substantial new analyses.

3. In disagreement with one point from Reviewer 3, I believe it is fully appropriate to cite BioRxiv studies as evidence or motivation for current work. The idea that peer review constitutes a magic stamp of authenticity is incorrect. I encourage the authors to cite relevant work that has been made publicly available, even in the absence of formal publication in a traditional journal.

Reviewer #2 (Remarks to the Author):

The revised manuscript addresses my concerns. The use of mixed-effects models to account for nested structure of the data is a great addition. I still find the proposed interpretation of the differences between stability of grating and movie responses somewhat unlikely (see point 1 below). Speculation about potential relationship between local circuit organization and representational stability is welcome in the discussion. However, I do not think that the final sentence of the abstract (that the results are "related to differences in preexisting circuit architecture of co-tuned neurons") is sufficiently grounded in evidence.

This point regarding interpretation notwithstanding, I think this is an interesting and well-executed study. Well done! My major comments below are organized around the same 5 points as in my original report.

1. I am still not entirely convinced by the argument that reciprocal connectivity of neurons co-tuned for the same grating might constrain representational drift compared to movie stimuli. It is true that natural movie responses tend to be sparser than grating responses, but among neurons co-activated by a given grating stimulus only the subset that shares selectivity for other stimulus features are likely to be strongly connected.

It seems more plausible that grating stimuli underestimate representational drift because they only probe one (or a small number of) stimulus dimension. Consider representational drift as a random walk through feature space and RDI as a function of the distance traversed in that space. Grating stimuli only probe one dimension of that space, while movies probe many dimensions. The length of a 1D random vector will always be shorter than that of a high-dimensional vector with the same variance for each dimension.

Related to this point, I find the schematic in Figure 6e quite confusing. Actual connectivity would be the same for PGD and MOV panels. What is meant by functional connectivity in this context? Would it not be a difference in stimulus tuning? It is not clear how the stimulus tuning for MOV stimuli might be illustrated with a single color scale.

2. Great additions. I think the additional data in Supplemental Figure 11 nicely illustrate that pupil size and arousal are very unlikely to explain differences in RDI between gratings and movies.

3. Thank you for the clarification. It seems only very specific changes would result in representational drift without changing signal correlations. If several neurons change their response at a certain timepoint in the movie, that would not affect signal correlations only if they respond solely at that time point. If they also respond at other time points and their response then is unchanged, the signal correlations will be altered. On the other hand, if neurons simply alter their gain and change responses at all time points by the same scale factor, this will neither affect signal correlations nor cause representational drift as measured by RDI.

4. From the new data in figure S8, it is clear that repeated stimulus presentation cannot fully explain representational drift.

5. Another great addition. It is clear that age makes only a minor contribution to variability in RDI and that RDI is higher for movie stimuli across the range of animal ages used in the manuscript.

Minor points:

1. I am confused by the example pupil images in Supplemental Figure 11. During two-photon imaging, the pupil typically appears back-illuminated by scattered laser light but in the examples the pupil appears dark. Based on the methods (lines 603-607), it does not appear that the camera was equipped with a filter that would remove the laser light. So why is the pupil black?

2. Line 484: "since there are numerous features pr" – looks like some text may be missing.

3. Lines 642-643: I am assuming magnification here refers to the zoom setting in the acquisition software. This is not a very useful number as the conversion between the zoom factor and scan angle is somewhat arbitrary and magnification is not quite the right term in this context. I think reporting the size of the FOV in microns or the spatial scale of imaging pixels is sufficient and more useful.

Reviewer #3 (Remarks to the Author):

Overall, the authors did a nice job addressing the reviewer concerns. The additional controls for arousal level/pupil diameter are convincing, and including the additional technical details on acquisition are useful.

Major

Regarding the representational drift analysis across stimuli for inhibitory neurons sharply- and broadly tuned for orientation, while I appreciate that this functional classification cannot be mapped to inhibitory cell types defined at the molecular level, the data are of interest. As another reviewer noted, the manuscript does not go into great depth regarding mechanism. Given that, a complete/ thorough description at a phenomenological level is of value. The difference between MOV sharp and MOV broad is striking; including these data will likely increase impact of the work. From this result, the authors are justified to conclude that RDI is not uniform across inhibitory neurons. ...note, not all PV neurons are broadly tuned, so in fact this functional classification may not map onto PV, SOM, etc.

Minor

Ref #37 has typos

Reviewer #1 (Remarks to the Author):

1. Prior work on variable sensory responses over time was not limited to perceptual learning. Puscian et al. 2020 showed that passive presentation of visual stimuli in the absence of any task was sufficient to drive significant reduction in visual responses, with ~70 stimuli per day. Recent work from the Axel lab also demonstrates in the olfactory system that passive stimulus presentation is associated with representational drift. These earlier findings should be emphasized in the introduction to not give the sense that the present manuscript is the first to show such longitudinal variation.

We did not intend to give the impression that our manuscript is the first to show longitudinal variation. We have added the Schoonover et al. 2021 paper (not yet published at the time of the previous submission), as well as the Puscian et al. 2020 and Deitch et al. papers to our discussion of previous literature on longitudinal imaging.

2. The authors have done a reasonable job responding to many of the issues I raised (some in common with other reviewers). My major outstanding concern, is what the RDI is really measuring. It is unlike other standard metrics (peak response amplitude or integral, tuning properties, accurate representation of the stimulus, etc). It seems to me that the correlation in activity during stimulus presentation across trials/sessions (i.e., the RDI) has the potential to be confounded by changes in response amplitude, kinetics, and relative weight in comparison to ongoing stimulus-unrelated activity. None of these varying contributors are really fleshed out here, leaving me unsure how to interpret the results. What does a change in RDI "mean" other than a reduction in correlation of signal? Is it more difficult to predict that a stimulus occurred? Is it more difficult to predict features of the stimulus from the recorded activity? I believe the authors need to do a much more thorough job working through these ideas in the discussion at least, in the absence of substantial new analyses.

The reason we use a correlation-based measure is that responses to natural movies consist of multiple response peaks of varying amplitude, and cannot be captured by measuring the amplitude/integral or by a standard tuning curve. As a result, we cannot use the metrics used by other groups (e.g., Schoonover et al., 2021) that analyzed responses to simpler stimuli. Since we do not know a priori which feature of the movie the neuron is responding to, we use the within-session correlation to determine the robustness of the response to the stimulus, as has been used for within-session measurements in a number of previous papers (Kampa et al., 2011; Froudarakis et al., 2014; Chen et al., 2015; Rikhye et al., 2015), and the between-session correlation to determine the robustness of the signal across sessions. Note that the only other manuscript that has analyzed responses to natural movie stimuli across sessions (Deitch et al., BioRxiv) used a similar correlation-based metric (population vector correlation). We have added additional detail on our selection of the RDI metric (line 155-161), but in this case we believe that the use of this measure is well justified. In addition, we also have considerable analysis of the changes using a metric that is not correlation based in Figure 2, where we investigate the changes in amplitude that occur on an event-by-event basis. The results of this analysis were very similar to the RDI metric used elsewhere in the paper.

As to the meaning of the change in RDI, it would certainly alter the ability of individual neurons to encode specific stimulus features across time. However, the change in population representation is more complicated, since a downstream decoder can take advantage of redundant population-level representations, or plasticity in the read out (Rule et al., 2020). We

are collaborating with a theory group to explore this further in a separate manuscript. We have added additional discussion of this point to the manuscript (line 487-489).

3. In disagreement with one point from Reviewer 3, I believe it is fully appropriate to cite BioRxiv studies as evidence or motivation for current work. The idea that peer review constitutes a magic stamp of authenticity is incorrect. I encourage the authors to cite relevant work that has been made publicly available, even in the absence of formal publication in a traditional journal.

We are inclined to agree, and have added this reference back to the introduction (line 78).

Reviewer #2 (Remarks to the Author):

The revised manuscript addresses my concerns. The use of mixed-effects models to account for nested structure of the data is a great addition. I still find the proposed interpretation of the differences between stability of grating and movie responses somewhat unlikely (see point 1 below). Speculation about potential relationship between local circuit organization and representational stability is welcome in the discussion. However, I do not think that the final sentence of the abstract (that the results are “related to differences in preexisting circuit architecture of co-tuned neurons”) is sufficiently grounded in evidence.

This point regarding interpretation notwithstanding, I think this is an interesting and well-executed study. Well done! My major comments below are organized around the same 5 points as in my original report.

We thank the reviewer for their kind comments. Given the lack of definitive evidence for our circuit organization model of differential stability, we have softened the claim in the final line of the abstract.

1. I am still not entirely convinced by the argument that reciprocal connectivity of neurons co-tuned for the same grating might constrain representational drift compared to movie stimuli. It is true that natural movie responses tend to be sparser than grating responses, but among neurons co-activated by a given grating stimulus only the subset that shares selectivity for other stimulus features are likely to be strongly connected.

It seems more plausible that grating stimuli underestimate representational drift because they only probe one (or a small number of) stimulus dimension. Consider representational drift as a random walk through feature space and RDI as a function of the distance traversed in that space. Grating stimuli only probe one dimension of that space, while movies probe many dimensions. The length of a 1D random vector will always be shorter than that of a high-dimensional vector with the same variance for each dimension.

Related to this point, I find the schematic in Figure 6e quite confusing. Actual connectivity would be the same for PGD and MOV panels. What is meant by functional connectivity in this context? Would it not be a difference in stimulus tuning? It is not clear how the stimulus tuning for MOV stimuli might be illustrated with a single color scale.

The reasoning is that neurons co-activated by the gratings will generally all share the same orientation preference, and are much more likely to be connected in reciprocal subnetworks (Ko et al., 2011), which will help preserve the tuning. On the other hand, neurons coactivated during a particular frame of a movie might be responding to very different features. For example, one neuron might be responding to the spatial frequency of the stimulus within the frame while another might be responding to its temporal frequency. As a result, they are not very likely to be connected into reciprocal subnetworks, and thereby less likely to be constrained to a stable response.

The reviewer's comment about representational drift as a function of dimensionality is an interesting one. However, there are several reasons we do not think it fully explains our data. First, the phase scrambling in Figure 5 significantly reduces the dimensionality of the natural scenes (by eliminating higher order spatial and temporal correlations), but had no effect on the amount of representational drift. Further, in other sensory systems (and hippocampus), considerable drift has been observed even in relatively low dimensional feature spaces (Ziv et al., 2013; Driscoll et al., 2017; Schoonover et al., 2021). This is not to say that stimulus dimensionality has no effect on representational drift, but we do not think it can fully explain the data.

Finally, we realized our schematic in Figure 6e was unintentionally misleading since we used the same positions for the neurons responding to PDG and MOV, implying it was the same cells in each case. It should be noted that these would not be the same cells (since neurons iso-tuned for orientation would be unlikely to respond to the same frame of the movie). We have changed the figure to make this clear. As far as the color scale, this is just a schematized manner of showing tuning to similar frames. To be fully accurate, each neuron would have multiple colors, but we worry this would complicate the figure and obscure the point. The goal of the figure is simply to help the readers visualize the argument about the potential role of reciprocal connectivity in constraining drift. We have added additional text to explain this figure and the general argument (including the example above), since we agree that it was not sufficiently clear in the previous version of the manuscript (line 494-504).

2. Great additions. I think the additional data in Supplemental Figure 11 nicely illustrate that pupil size and arousal are very unlikely to explain differences in RDI between gratings and movies.

Thank you.

3. Thank you for the clarification. It seems only very specific changes would result in representational drift without changing signal correlations. If several neurons change their response at a certain timepoint in the movie, that would not affect signal correlations only if they respond solely at that time point. If they also respond at other time points and their response then is unchanged, the signal correlations will be altered. On the other hand, if neurons simply alter their gain and change responses at all time points by the same scale factor, this will neither affect signal correlations nor cause representational drift as measured by RDI.

Agreed.

4. From the new data in figure S8, it is clear that repeated stimulus presentation cannot fully explain representational drift.

Agreed.

5. Another great addition. It is clear that age makes only a minor contribution to variability in RDI and that RDI is higher for movie stimuli across the range of animal ages used in the manuscript.

Thank you.

Minor points:

1. I am confused by the example pupil images in Supplemental Figure 11. During two-photon imaging, the pupil typically appears back-illuminated by scattered laser light but in the examples the pupil appears dark. Based on the methods (lines 603-607), it does not appear that the camera was equipped with a filter that would remove the laser light. So why is the pupil black?

Thank you for noting that, we actually used an external 850 nm IR light source for visualizing the eye - your comment made us realize that the light source was not included in the methods. We have corrected that omission (line 626).

We did not see back-illumination of the pupil during 2P imaging. We suspect either because our light source was significantly brighter at the surface of the eye than the scattered laser or because the quantum efficiency of our camera is significantly higher at 850 nm than 920 nm (our 2P excitation frequency).

2. Line 484: "since there are numerous features pr" – looks like some text may be missing.

Fixed, thank you.

3. Lines 642-643: I am assuming magnification here refers to the zoom setting in the acquisition software. This is not a very useful number as the conversion between the zoom factor and scan angle is somewhat arbitrary and magnification is not quite the right term in this context. I think reporting the size of the FOV in microns or the spatial scale of imaging pixels is sufficient and more useful.

Agreed, we have changed our description of the imaging frames to include the number of pixels and the size of the field in microns.

Reviewer #3 (Remarks to the Author):

Overall, the authors did a nice job addressing the reviewer concerns. The additional controls for arousal level/pupil diameter are convincing, and including the additional technical details on acquisition are useful.

Thank you.

Major

Regarding the representational drift analysis across stimuli for inhibitory neurons sharply- and broadly tuned for orientation, while I appreciate that this functional classification cannot be

mapped to inhibitory cell types defined at the molecular level, the data are of interest. As another reviewer noted, the manuscript does not go into great depth regarding mechanism. Given that, a complete/ thorough description at a phenomenological level is of value. The difference between MOV sharp and MOV broad is striking; including these data will likely increase impact of the work. From this result, the authors are justified to conclude that RDI is not uniform across inhibitory neurons. ...note, not all PV neurons are broadly tuned, so in fact this functional classification may not map onto PV, SOM, etc.

Agreed, we have included the sharp/broad interneuron data as a supplementary figure (Supplementary Figure 13) and have described the results in the text (line 355-357).

Minor
Ref #37 has typos

Thank you for catching that (odd) typo, we have corrected it.